# Wan-Move: Motion-controllable Video Generation via Latent Trajectory Guidance

**Ruihang Chu**[1,2*†‡]   **Yefei He**[1*]   **Zhekai Chen**[3*]   **Shiwei Zhang**[1†]   **Xiaogang Xu**[4]
**Bin Xia**[4]   **Dingdong Wang**[4]   **Hongwei Yi**   **Xihui Liu**[3]   **Hengshuang Zhao**[3]
**Yu Liu**[1]   **Yingya Zhang**[1]   **Yujiu Yang**[2†]

[1]Tongyi Lab, Alibaba Group   [2]Tsinghua University   [3]HKU   [4]CUHK

Github: https://github.com/ali-vilab/Wan-Move

## Abstract

We present Wan-Move, a simple and scalable framework that brings motion control to video generative models. Existing motion-controllable methods typically suffer from coarse control granularity and limited scalability, leaving their outputs insufficient for practical use. We narrow this gap by achieving precise and high-quality motion control. Our core idea is to directly make the original condition features motion-aware for guiding video synthesis. To this end, we first represent object motions with dense point trajectories, allowing fine-grained control over the scene. We then project these trajectories into latent space and propagate the first frame's features along each trajectory, producing an aligned spatiotemporal feature map that tells how each scene element should move. This feature map serves as the updated latent condition, which is naturally integrated into the off-the-shelf image-to-video model, *e.g.*, Wan-I2V-14B, as motion guidance without any architecture change. It removes the need for auxiliary motion encoders and makes fine-tuning base models easily scalable. Through scaled training, Wan-Move generates 5-second, 480p videos whose motion controllability rivals Kling 1.5 Pro's commercial Motion Brush, as indicated by user studies. To support comprehensive evaluation, we further design MoveBench, a rigorously curated benchmark featuring diverse content categories and hybrid-verified annotations. It is distinguished by larger data volume, longer video durations, and high-quality motion annotations. Extensive experiments on MoveBench and the public dataset consistently show Wan-Move's superior motion quality. Code, models, and benchmark data are made available.

## 1   Introduction

Motion lies at the heart of video generation as it fundamentally transforms static images into dynamic visual narratives. Recognizing its importance, both the research community [1, 2, 3] and commercial players [4, 5, 6] have devoted considerable effort to controlling motion in video generative models.

The essence of motion control lies in injecting a motion guidance signal into the video generation process. Thus, the two key choices are (i) how to represent the guidance signal and (ii) how to integrate it into the generator. First, existing motion guidance representations can be broadly classified into sparse and dense types. Sparse representations include bounding boxes [7, 8] and segmentation masks [1, 9, 10]. Although these signals can steer an object's global movement, they fail to control

---

[*]Equal contribution

[†]Corresponding authors

[‡]Project leader

39th Conference on Neural Information Processing Systems (NeurIPS 2025).

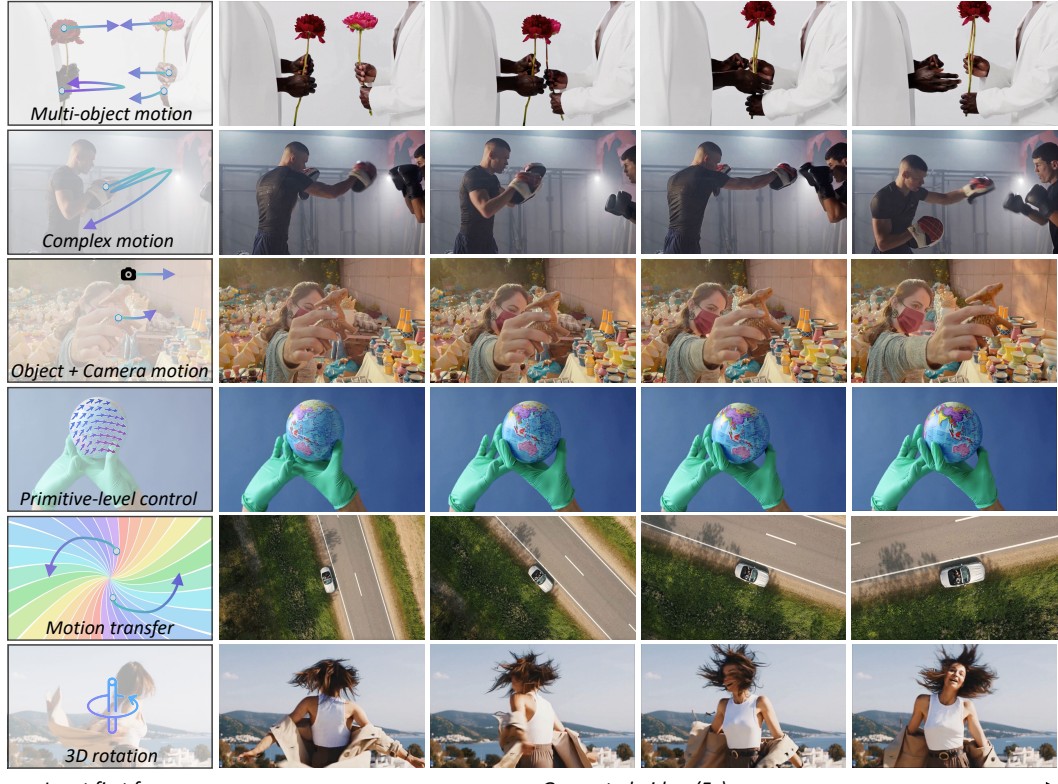

Figure 1: Wan-Move is a image-to-video generation framework that supports diverse motion control applications. The generated samples (832×480p, 5s) exhibits high visual fidelity and accurate motion.

local motions. In contrast, dense representations, such as pixel-wise optical flow [2, 11, 12, 3] and point trajectories [13, 14], offer more fine-grained control ability. Yet, optical flow requires an additional model for flow estimation during inference, which adds cumulative errors across frames and hampers scalability. While point trajectories are easy to specify during inference, each track is only a single-pixel thread and lacks surrounding spatial context. This makes it hard to align textures and motion patterns across neighboring regions. Second, to inject the guidance signal into generative models, a range of motion encoders have been designed [14, 15, 1, 13, 16, 17], with ControlNet [18] being a popular way to fuse motion cues. However, all of these pipelines introduce extra motion-aware modules, which may degrade the motion signal during processing and make it harder to fine-tune the video-generation backbone at scale.

To tackle these challenges, we present Wan-Move, a novel motion-control framework that builds on the existing image-to-video (I2V) generation model without adding auxiliary motion-processing modules. Our core idea is to inject motion information by directly editing the image condition features. We turn it into an updated latent guide that conveys both appearance and motion throughout video generation. Thanks to this simple and effective design, Wan-Move delivers high-quality motion control and scales easily by fine-tuning the powerful I2V backbone.

Specifically, we represent motion trajectories using point tracks [13] as they capture fine-grained local and global movement. Unlike prior work [13] that embeds point trajectories into latent features, we transfer each trajectory from pixel space into latent coordinates. As I2V generation aims to animate the first frame, we guide this process by copying the first-frame feature at each tracked position to its corresponding location in later frames along the latent trajectory. Each copied feature preserves rich context, thus the propagated signal drives more natural local motion, as verified in Sec. 5.3. Moreover, since motion guidance is injected by editing the image condition features, we add no extra modules. As a result, Wan-Move can plug straight into the I2V backbone, such as Wan-I2V-14B [19], and support scalable fine-tuning with fast convergence. Fig. 1 shows that Wan-Move generates high-fidelity video clips (832×480p, 5s) with precise motion control, enabling a diverse set of applications, as illustrated in Sec. 5.4. To our best knowledge, it is the first research model (*to be open-sourced*) to match the visual quality of commercial products such as Kling 1.5 Pro's Motion Brush [4].

To set a rigorous, comprehensive evaluation for motion-control methods, we introduce a free-license benchmark termed MoveBench. Compared with existing benchmarks [20, 21, 22] that offer fewer clips, shorter durations, and incomplete motion annotations, MoveBench provides more data, greater diversity, and reliable motion annotations (Fig. 5). Concretely, we design a curation pipeline to categorize the video library into 54 content categories, 10-25 videos each, giving rise to over 1000 cases to ensure a broad scenario coverage. All video clips maintain a 5-second duration to facilitate evaluation of long-range dynamics. Every clip is paired with detailed motion annotations for single or multiple objects. They include both precise point trajectories and sparse segmentation masks to fit a wide range of motion-control models. We ensure annotation quality by developing an interactive labeling pipeline. It combines human labeling with SAM [23] predictions, marrying annotation precision with automated scalability. In summary, our contributions are as follows:

- We propose Wan-Move for motion control in image-to-video generation. Unlike prior approaches that require motion encoding, it injects the motion guidance by editing condition features, adding no new modules and allowing the easy fine-tuning of base models at scale.

- We introduce MoveBench, a comprehensive and well-curated benchmark to assess motion control. A hybrid human+SAM labeling pipeline ensures annotation quality.

- Extensive experiments on MoveBench and public datasets show that Wan-Move supports diverse motion-control tasks and delivers commercial-grade results with scaled training.

## 2   Related Work

**Video generation models.** Video diffusion models [24] pioneer the extension of denoising diffusion probabilistic models (DDPMs) to video generation through a 3D U-Net architecture. Subsequent advancements, such as Imagen Video [25] and Phenaki [26], enhance this framework to produce longer and higher-resolution sequences. Nevertheless, these CNN-based approaches [27, 28, 29] face limitations in capturing long-range spatiotemporal dependencies. Transformer-based architectures [30, 31, 32, 33, 34, 35, 36, 37, 38, 39, 40, 41, 42, 43, 44, 45, 46] overcome this bottleneck and greatly improve training scalability. Recent innovations, including CogVideoX [47] and HunyuanVideo [6], further validate the efficacy of spatio-temporal attention mechanisms for coherent video synthesis. Notably, Wan [19] introduces an efficient framework for both text-to-video and image-to-video generation, setting a new standard for open-source video models. Our Wan-Move, introduces how to leverage latent trajectory guidance to enable motion control upon the image-to-video diffusion model, enabling precise motion control while preserving visual fidelity.

**Motion-controllable video generation.** To adapt pretrained video generation models for motion-controllable synthesis, training-free methods [48, 49, 50, 51] optimize input noisy latents or manipulate attention mechanisms, enabling zero-shot control. However, these approaches often exhibit performance degradation when controlling fine-grained or multi-object motion. In contrast, fine-tuning based methods [52, 2, 15, 1, 3, 14, 20, 53, 16, 54, 55, 56, 10, 12, 57, 58, 59] leverage diverse motion signals and introduce various techniques to integrate them into the base model. While these methods significantly enhance output quality, they typically require auxiliary encoders or fusion modules, complicating the model architecture and limiting training scalability. Among these studies, the most relevant to our work is Motion Prompting [13], as both employ point trajectories to represent motion guidance. However, we differ in two key aspects. First, Motion Prompting [13] encodes point tracks via random embeddings in pixel space, where the guidance is pixel-level threads that lack surrounding context to offer local control. We express point trajectories in latent space using the image feature, providing rich local information and finer control. Second, Motion Prompting [13] integrates motion guidance through a separate ControlNet [18], whereas we directly use the pretrained base model without architectural modifications, which facilitates its scalable fine-tuning. Sec. 5.3 provides quantitative and qualitative evidence of our advantages.

For more specific robotic scenarios, works [60, 61] rely on pretrained DINOv2 features [62] to transfer object representations across frames for motion control in generated videos, yet both delivers DINOv2's limitation in representing motion signals. DINOv2 excels at high-level semantic encoding but lacks fine-grained object details. Thus, GEM [1] employs additional identity embeddings to distinguish objects and train an ObjectNet to bridge the domain gap between DINOv2 and the UNet's feature space. Moreover, the 14 patch size in DINOv2 may restrict the granularity of the proposed

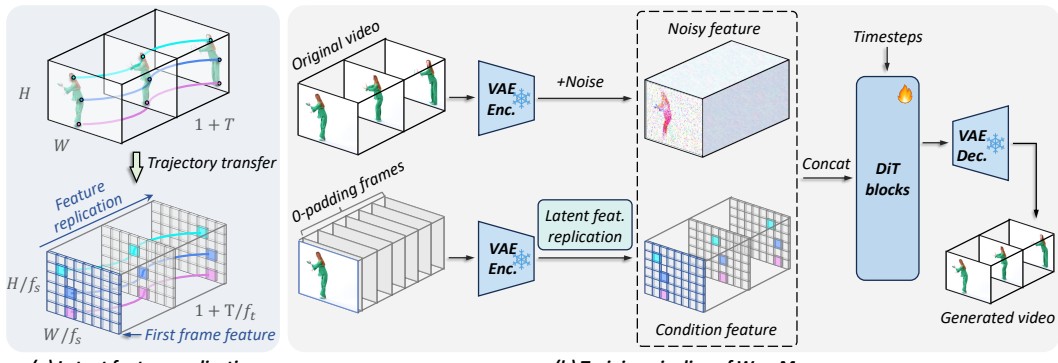

(a) Latent feature replication      (b) Training pipeline of Wan-Move

Figure 2: (a) To inject motion guidance, we transfer point trajectories from videos to latent space, then replicate the first frame feature into subsequent zero-padded frames along each latent trajectory. (b) Wan-Move is trained upon an existing image-to-video generation model (*e.g.*, work [19, 29]), with an efficient latent feature replication step (as in (a)) to update the condition feature. The CLIP [65] image encoder and umT5 [66] text encoder from the base model are omitted for simplicity.

control. In contrast, our approach mitigates these limitations by employing native VAE in I2V foundation models without relying on any auxiliary modules (*e.g.*, identity embeddings).

**Benchmark for motion-controllable video generation.** Most motion-controllable work evaluates on small, task-specific datasets, which are typically ad hoc, limited in scope, or insufficient for evaluating long-range dynamics and multi-object interactions. For example, datasets such as DAVIS [21] and VIPSeg [22] have been repurposed for trajectory control methods, yet their short clip durations and sparse annotations make them inadequate for assessing long-term consistency or complex interactions. While MagicBench [20] expands to 600 video clips, it categorizes samples solely by object count and relies on automatically generated labels from noisy pipelines, limiting the annotation precision. To address these limitations, we introduce MoveBench, a comprehensive benchmark for motion-controllable video generation. It includes carefully selected 1018 videos with extensive annotations, long-range dynamics, and 54 well-classified content patterns.

## 3 Method

### 3.1 Preliminary

Video diffusion models [24, 19, 6, 63] apply Gaussian noise to clean data during the forward process and learn a reverse process to denoise and generate videos. To reduce computational costs, the denoising network typically operates on latent video representations obtained from a pretrained VAE [64]. Given an input video $\mathbf{V} \in \mathbb{R}^{(1+T) \times H \times W \times 3}$, the encoder $\mathcal{E}$ compresses both the temporal and spatial dimensions with compression ratios $f_t$ (temporal) and $f_s$ (spatial), while expanding the channel dimension to $C$, yielding $\mathbf{x} = \mathcal{E}(\mathbf{V}) \in \mathbb{R}^{(1+\frac{T}{f_t}) \times \frac{H}{f_s} \times \frac{W}{f_s} \times C}$. The decoder $\mathcal{D}$ then reconstructs the video from the latent representation as $\hat{\mathbf{V}} = \mathcal{D}(\mathbf{x})$.

Our work focuses on motion-controllable image-to-video (I2V) generation, where models are required to generate motion-coherent videos based on the input first-frame image and motion trajectories. While the first frame will be encoded into the condition feature $\mathbf{z}_{\text{image}}$ by the VAE, motion trajectories, which can be represented in diverse formats, remain in pixel space. Thus, the key challenge lies in effectively encoding motion trajectories into the condition feature $\mathbf{z}_{\text{motion}}$ and injecting it into the generative model. To avoid the signal degradation and training difficulties associated with additional motion encoder and fusion modules, we aim to develop a motion-control framework that leverages existing I2V models without architectural modifications, as detailed below.

### 3.2 Latent Trajectory Guidance

To enable video generation conditioned on the first frame, an effective approach of popular I2V models [19, 29] concatenates the latent noise $\mathbf{x}_t$ and the first-frame condition feature $\mathbf{z}_{\text{image}}$ along

the channel dimension. $\mathbf{z}_{\text{image}}$ is obtained by encoding the first frame $\mathbf{I}$ along with zero-padded subsequent frames $\mathbf{0}_{T \times H \times W \times C}$ using a pretrained VAE encoder $\mathcal{E}$:

$$\mathbf{z}_{\text{image}} = \mathcal{E}\left(\text{concat}\left[\mathbf{I}, \mathbf{0}_{T \times H \times W \times 3}\right]\right) \in \mathbb{R}^{(1+\frac{T}{f_t}) \times \frac{H}{f_s} \times \frac{W}{f_s} \times C}. \tag{1}$$

For motion guidance representation, we adopt point trajectories, following prior studies [13, 14], as they provide fine-grained control and capture both local and global motion. Formally, a point trajectory of length $1 + T$ can be represented as $\mathbf{p} \in \mathbb{R}^{(1+T) \times 2}$, where $\mathbf{p}[n] = (x_n, y_n)$ specifies the trajectory location in the $n$-th frame in the pixel space. Existing methods often employ auxiliary modules to encode and integrate trajectories into the backbone. Yet, this approach may degrade motion signals during motion encoding. In addition, training extra modules increases the complexity of fine-tuning the base model at scale. This raises a key question: *Can we inject pixel-space motion guidance without auxiliary modules?*

Intuitively, I2V generation aims to animate the first frame, while motion trajectories specify object positions in each subsequent frame. Given the translation equivariance of VAE models, latent features at corresponding trajectory positions should closely resemble those in the first frame. Motivated by this, we propose encoding trajectories directly into latent space via spatial mapping, eliminating the need for an extra motion encoder:

$$\tilde{\mathbf{p}}[n] = \begin{cases} \frac{\mathbf{p}[n]}{f_s} & \text{if } n = 0, \\ \frac{\sum_{i=(n-1) \cdot f_t + 1}^{n \cdot f_t} \mathbf{p}[i]}{f_t \cdot f_s} & 1 \leq n \leq \frac{T}{f_t}, \end{cases} \tag{2}$$

The latent trajectory position at the first frame is derived by spatial mapping, while for subsequent frames, it is averaged over each consecutive $f_t$ frames. This deterministically transforms pixel-space trajectories into latent space. To inject the obtained latent trajectories, we extract the latent features of the first frame at the initial trajectory point $\tilde{\mathbf{p}}[0]$ and replicate them across subsequent frames according to $\tilde{\mathbf{p}}$, leveraging the translation equivariance of latent features, as shown in Fig. 2 (a):

$$\mathbf{z}_{\text{image}}\left[n, \tilde{\mathbf{p}}[n, 0], \tilde{\mathbf{p}}[n, 1], :\right] = \mathbf{z}_{\text{image}}\left[0, \tilde{\mathbf{p}}[0, 0], \tilde{\mathbf{p}}[0, 1], :\right] \quad \text{for} \quad n = 1, \ldots, \frac{T}{f_t}. \tag{3}$$

Here, $\mathbf{z}_{\text{image}}[t, h, w, :]$ denotes the feature vector at temporal index $t$, height $h$, and width $w$. This operation efficiently injects motion guidance into the condition feature by updating $\mathbf{z}_{\text{image}}$, eliminating the need for explicit motion condition features and injection modules. An overview of the Wan-Move generation framework is presented in Fig. 2(b). When multiple visible point trajectories coincide at a given spatiotemporal position, we randomly select one trajectory's corresponding first-frame feature.

### 3.3 Training and Inference

**Training data.** We curate a high-quality training dataset, which undergoes rigorous two-stage filtering to ensure both visual quality and motion consistency. First, we manually annotate the visual quality of 1,000 samples and use them to train an expert scoring model for initial quality assessment. To further enhance temporal coherence, we introduce a motion quality filtering stage. Specifically, for each video, we extract SigLIP [67] features from the first frame and compute the mean SigLIP features for the remaining frames. The cosine similarity between these features serves as our stability metric. Based on empirical analysis of 10,000 samples, we establish a threshold to retain only videos where the content remains consistent with the initial frame. This two-stage pipeline produces a final dataset of 2 million high-quality 720p videos with strong visual quality and motion coherence. Additional details on the training data sources are provided in the supplementary material.

**Modeling training.** Based on our training dataset, we use CoTracker [68] to track the trajectories of a dense 32×32 grid of points. For each training iteration, we sample $k$ trajectories from a mixed distribution: with 5% probability, no trajectory is used ($k = 0$); with 95% probability, $k$ is uniformly sampled from 1 to 200. Notably, we retain a 5% probability of dropping motion conditions, which effectively preserves the model's original image-to-video generation capability. For the selected trajectories, we extract the first-frame features and replicate them to subsequent zero-padded frames, as formalized by Eq. (3). Since CoTracker distinguishes between visible and

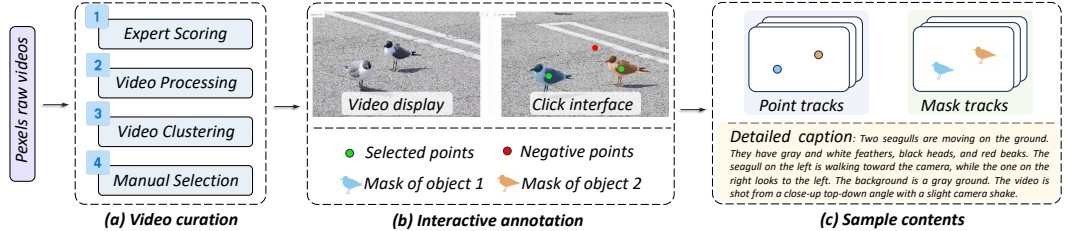

Figure 3: Construction pipeline of MoveBench to obtain high-quality samples with rich annotations.

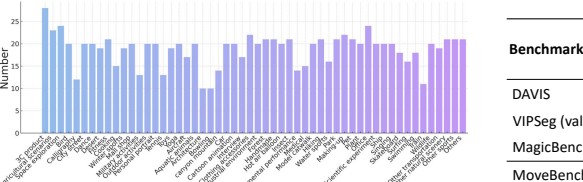

Figure 4: Balanced sample number per class.

| Benchmark | Statistics | | | Trajectory annotations | |
|---|---|---|---|---|---|
| | Videos | Frames | Video categorization | Mask track | Point track |
| DAVIS | 50 | 35-100 | ✗ | ✓ | ✗ |
| VIPSeg (val) | 343 | 24 | ✗ | ✓ | ✗ |
| MagicBench | 600 | 49 | ✗ | ✓ | ✗ |
| MoveBench | **1018** | **81** | ✓ | ✓ | ✓ |

Figure 5: Comparison with related benchmarks.

occluded point trajectories, we perform feature replication only along the visible trajectories. During training, the model parameters $\theta$ is initialized from the I2V model [19] and fine-tuned to predict the vector field $\mathbf{v}_t(\mathbf{x}_t)$ that transports samples from the noise distribution to the data distribution [69]:

$$\mathcal{L}_{\text{FM}}(\theta) = \mathbb{E}_{t,\mathbf{x}_t,\mathbf{c}} \left[ \|\mathbf{v}_\theta(\mathbf{x}_t, t, \mathbf{c}) - \mathbf{v}_t(\mathbf{x}_t)\|^2 \right], \tag{4}$$

where $\mathbf{c}$ denotes the union of the generation condition.

**Inference with Wan-Move.** The inference process closely resembles the original I2V model, with an additional latent feature replication operation. Specifically, Wan-Move conditions generation on three inputs: (1) a text prompt, (2) an input image as the first frame, and (3) sparse or dense point trajectories for motion control. Pretrained umT5 [66] and CLIP [65] models are employed to encode global context from the text prompt and first frame, respectively. The resulting image embedding $\mathbf{z}_{\text{global}}$ and text embeddings $\mathbf{z}_{\text{text}}$ are then injected into the DiT backbone via decoupled cross-attention [18]. Additionally, a VAE is used to extract the first-frame condition feature $\mathbf{z}_{\text{image}}$, which will be injected through latent feature replication (as detailed in Sec. 3.2). Classifier-free guidance is applied to enhance alignment with conditional information. Formally, let unconditional vector field $\mathbf{v}_{\text{uncond}} = \mathbf{v}_\theta(\mathbf{x}_t, t, \mathbf{z}_{\text{image}}, \mathbf{z}_{\text{global}})$, and conditional vector field $\mathbf{v}_{\text{cond}} = \mathbf{v}_\theta(\mathbf{x}_t, t, \mathbf{z}_{\text{image}}, \mathbf{z}_{\text{global}}, \mathbf{z}_{\text{text}})$. The guided vector field $\tilde{\mathbf{v}}_\theta(\mathbf{x}_t, t, \mathbf{z}_{\text{image}}, \mathbf{z}_{\text{global}}, \mathbf{z}_{\text{text}})$ is a weighted combination of the conditional and unconditional outputs, with the guidance scale $w$:

$$\tilde{\mathbf{v}}_\theta(\mathbf{x}_t, t, \mathbf{z}_{\text{image}}, \mathbf{z}_{\text{global}}, \mathbf{z}_{\text{text}}) = \mathbf{v}_{\text{uncond}} + w(\mathbf{v}_{\text{cond}} - \mathbf{v}_{\text{uncond}}) \tag{5}$$

## 4 MoveBench

Current benchmarks for motion-controllable video generation suffer from small scale, short duration, and lack precise, comprehensive motion annotations, thus introducing bias and limiting granularity. To address these gaps, we introduce MoveBench, a high-quality benchmark with 1018 videos (480×832 resolution, 5-second duration), designed for comprehensive evaluation of motion-controllable generation, as illustrated in Fig. 3-5. The evaluation videos are selected from Pexels [70], a large-scale, high-quality dataset containing about 400K videos, all released under a free license.

MoveBench combines algorithmic curation with human expertise to ensure diverse, representative, and precisely annotated motion data. Compared to prior works, it offers three key features: **(i) High quality.** We curate videos through a rigorous four-stage pipeline, as illustrated in Fig. 3(a). We first utilize the expert scoring model obtained from Sec. 3.3 to score videos based on visual quality, filtering out low-quality content. Then, the selected videos are cropped to 480p and uniformly sampled to 81 frames to ensure temporal consistency. Finally, videos are clustered into 54 content categories and we manually select the 15-25 most representative examples for each category, balancing diversity and quality (Fig. 4). **(ii) Precise annotations.** As shown in Fig. 5, we provide both point and mask annotations, so that methods using mask guidance signals can also be evaluated using our benchmark.

Table 1: Performance comparisons on MoveBench and DAVIS. Wan-Move consistently yields substantial improvements in both visual fidelity and motion quality across all metrics.

| Method | MoveBench | | | | | DAVIS | | | | |
|---|---|---|---|---|---|---|---|---|---|---|
| | FID↓ | FVD↓ | PSNR↑ | SSIM↑ | EPE↓ | FID↓ | FVD↓ | PSNR↑ | SSIM↑ | EPE↓ |
| ImageConductor [53] | 34.5 | 424.0 | 13.4 | 0.49 | 15.6 | 54.2 | 513.6 | 11.6 | 0.47 | 14.8 |
| LeviTor [16] | 18.1 | 98.8 | 15.6 | 0.54 | 3.4 | 22.0 | 115.4 | 13.3 | 0.51 | 3.7 |
| Tora [14] | 22.5 | 100.4 | 15.7 | 0.55 | 3.3 | 25.9 | 129.2 | 13.7 | 0.49 | 3.5 |
| MagicMotion [20] | 17.5 | 96.7 | 14.9 | 0.56 | 3.2 | 24.2 | 113.4 | 12.8 | 0.53 | 3.5 |
| Wan-Move (Ours) | **12.2** | **83.5** | **17.8** | **0.64** | **2.6** | **14.7** | **94.3** | **16.5** | **0.61** | **2.5** |

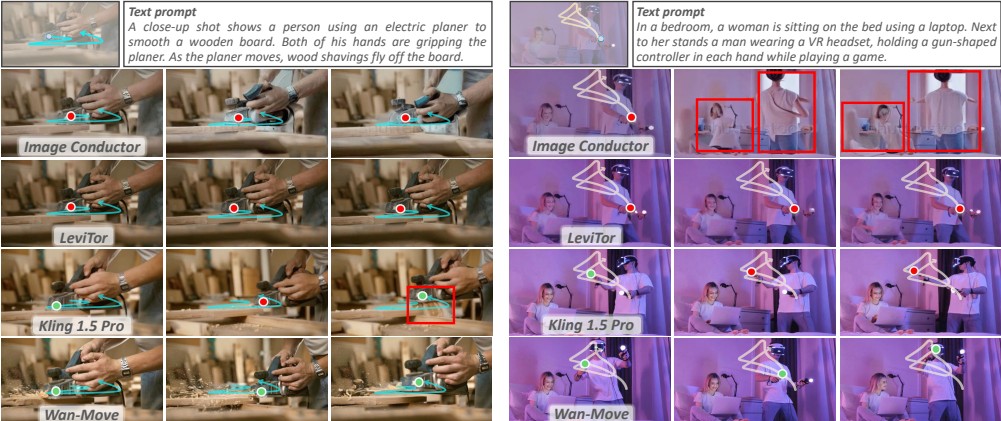

Figure 6: Qualitative comparisons between Wan-Move and recent approaches, including both academic methods [53, 16] and commercial solutions [4]. Motions that deviate from the specified trajectories and major visual artifacts are marked with red dots and boxes, respectively.

To precisely annotate motion regions, we design an interactive annotation interface as shown in Fig. 3. Annotators click on a target region in the first frame, prompting SAM [23] to generate an initial mask. When the mask exceeds the desired area, annotators add negative points to exclude irrelevant regions. This is critical for isolating articulated motions or small objects in cluttered scenes. After annotation, each video contains at least one point indicating a representative motion, with 192 videos additionally including multiple-object motion trajectories. **(iii) Detailed captions.** We use Gemini [71] to generate dense descriptions covering objects, actions, and camera dynamics. Unlike segmentation datasets like DAVIS [21], our captions are tailored for video generation tasks.

## 5 Experiment

### 5.1 Experimental Setup

Wan-Move is implemented on top of Wan-I2V-14B [19], a state-of-the-art image-to-video (I2V) generation model. As described in Sec. 3.3, we fine-tune Wan-Move on a high-quality dataset consisting of 2M high-quality videos. Only the DiT backbone is trainable, while the image and text encoders remain frozen. During inference, we use a classifier-free guidance scale $w$ of 5.0 unless otherwise specified. Detailed training configurations are provided in the supplementary material.

To quantitatively evaluate the fidelity of generated videos, we compute standard video quality metrics including FID [72], FVD [73], PSNR, and SSIM [74]. To assess motion accuracy, we measure the L2 distance between ground truth tracks and those estimated from generated videos, following [13] in denoting this metric as end-point error (EPE). All evaluations are performed at a resolution of 480p.

### 5.2 Main Results

**Single-object motion control.** We present an extensive comparison between Wan-Move and recent motion-controllable video generation methods [53, 16, 14, 20, 4]. Quantitative results on MoveBench and the public DAVIS [21] are shown in Table 1. Qualitative visualizations are presented in Fig. 6. Unlike other methods that rely on point tracks for motion guidance, MagicMotion [20] takes as input

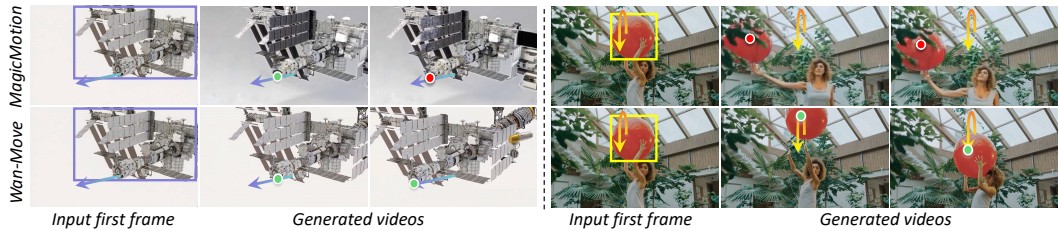

*Input first frame*     *Generated videos*     *Input first frame*     *Generated videos*

Figure 7: Qualitative comparisons with MagicMotion [20], controlling motions using sparse signals (*i.e.*, bounding boxes) as input. Motions that break the guidance are marked with red dots.

Table 2: MoveBench multi-object motion results.

| Method | FID↓ | FVD↓ | PSNR↑ | SSIM↑ | EPE↓ |
|---|---|---|---|---|---|
| ImageConductor | 77.5 | 764.5 | 13.9 | 0.51 | 9.8 |
| Tora | 53.2 | 350.0 | 14.5 | 0.54 | 3.5 |
| Wan-Move (Ours) | **28.8** | **226.3** | **16.7** | **0.62** | **2.2** |

Table 3: Our win rates in 2AFC human study.

| Method | Motion accuracy | Motion quality | Visual quality |
|---|---|---|---|
| LeviTor | 98.2 | 98.0 | 98.8 |
| Tora | 96.2 | 93.8 | 98.4 |
| MagicMotion | 89.4 | 96.4 | 98.2 |
| Kling 1.5 Pro | 47.8 | 53.4 | 50.2 |

Table 4: Ablation on motion guidance strategies.

| Motion guidance | FID↓ | FVD↓ | PSNR↑ | SSIM↑ | EPE↓ |
|---|---|---|---|---|---|
| Pixel replication | 17.3 | 91.0 | 15.3 | 0.56 | 3.7 |
| Random track embedding | 15.4 | 89.2 | 16.1 | 0.59 | 2.7 |
| Latent feature replication | **12.2** | **83.5** | **17.8** | **0.64** | **2.6** |

Table 5: Ablation on condition fusion strategies.

| Cond. fusion | FID↓ | FVD↓ | EPE↓ | Latency (s) |
|---|---|---|---|---|
| ControlNet | 12.4 | 84.6 | 2.5 | 987 (+225) |
| Concat. (Ours) | 12.2 | 83.5 | 2.6 | 765 (+3) |

sparse masks and bounding boxes. Since boxes can be directly derived from segmentation masks, they are inherently compatible with MoveBench that covering mask annotations. Hence, we also conduct comparisons using box-based inputs in Fig. 7. Among these methods, ImageConductor [53] exhibits poor performance in both image and motion quality, which can be attributed to its reliance on direct pixel-level trajectory injection, where single-pixel features lack sufficient semantic and texture information. The remaining methods report similar EPE (3.2–3.4), despite differing motion guidance approaches: Levitor [16] and MagicMotion [20] utilize the complex ControlNet [18], while Tora [14] adopts the lightweight adaLN [30]. Notably, our method achieves the **best motion control performance** (lowest EPE) and **video quality** (highest PSNR and SSIM) through latent trajectory replication without introducing additional parameters. This underscores the effectiveness of our latent trajectory guidance in adhering to motion constraints. Consistent results on the DAVIS dataset further validate the robustness of our approach.

**Multi-object motion control.** As MoveBench includes 192 cases with annotated multi-object motion, we further evaluate Wan-Move against baselines [53, 14] on this challenging setting, as presented in Table 2. Our method achieves significantly lower FVD and reduced EPE compared to other methods, highlighting its precise adherence to motion constraints in more complex scenarios.

**Human study.** We conduct a two-alternative forced-choice (2AFC) human evaluation comparing Wan-Move with SOTA approaches [14, 4, 16, 20]. Each method generated 50 conditioned samples, which are evaluated by 20 participants. The results, presented in Table 3, report Wan-Move's win rates across three metrics: motion accuracy, motion quality, and visual quality. Compared to Tora [14], Wan-Move achieves win rates exceeding 96% in all categories. When evaluated against the commercial model Kling 1.5 Pro, our method demonstrates competitive performance, with superior win rates in motion quality. This narrows the gap between research-oriented and commercial models.

## 5.3 Ablation Study

**Trajectory guidance strategy.** We investigate the impact of motion guidance strategies on video quality and motion consistency. Quantitative and qualitative results as presented in Table 4 and Fig. 8, respectively. Pixel replication applies pixel-level copy-paste along the original trajectory, followed by VAE encoding. Yet, since single-pixel features contain limited semantic and texture information, the resulting motion control is weak, as reflected by a high EPE value of 3.7 and generation

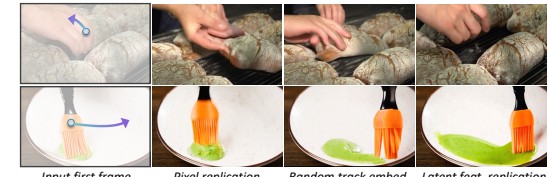

*Input first frame*    *Pixel replication*    *Random track embed.*    *Latent feat. replication*

Figure 8: Visualization of various guidance strategies.

Table 6: Ablation on maximum number of point trajectories (see Sec. 3.3) during training.

| Number | FID↓ | FVD↓ | PSNR↑ | SSIM↑ | EPE↓ |
|---|---|---|---|---|---|
| $N = 10$ | 12.8 | 86.6 | 17.6 | 0.62 | 3.3 |
| $N = 100$ | 12.9 | 84.7 | 17.7 | 0.65 | 2.7 |
| $N = 200$ | 12.2 | 83.5 | 17.8 | 0.64 | 2.6 |
| $N = 500$ | 13.3 | 83.9 | 17.6 | 0.63 | 3.0 |
| $N = 1024$ | 13.4 | 83.7 | 17.2 | 0.61 | 3.9 |

Table 7: Ablation on actual number of point trajectories during inference.

| Number | FID↓ | FVD↓ | PSNR↑ | SSIM↑ | EPE↓ |
|---|---|---|---|---|---|
| $N = 0$ | 12.8 | 87.9 | 17.9 | 0.64 | 12.4 |
| $N = 1$ | 12.2 | 83.5 | 17.8 | 0.64 | 2.6 |
| $N = 16$ | 10.6 | 78.3 | 18.2 | 0.67 | 2.2 |
| $N = 512$ | 7.7 | 51.0 | 20.3 | 0.75 | 1.5 |
| $N = 1024$ | 6.2 | 45.2 | 21.9 | 0.79 | 1.1 |

Table 8: Ablation on different I2V backbones and training data scale. Wan-Move attains better results under the same setting.

| Method | Backbone | Data scale | FID↓ | FVD↓ | PSNR↑ | SSIM↑ | EPE↓ |
|---|---|---|---|---|---|---|---|
| MagicMotion | CogVideoX-5B | 23K | 17.5 | 96.7 | 14.9 | 0.56 | 3.2 |
| Wan-Move-Cog-23K | CogVideoX-5B | 23K | 16.0 | 92.3 | 16.8 | 0.59 | 2.8 |
| Tora | CogVideoX-5B | 630K | 22.5 | 100.4 | 15.7 | 0.55 | 3.3 |
| Wan-Move-Cog-630K | CogVideoX-5B | 630K | 14.1 | 87.3 | 17.2 | 0.61 | 2.8 |
| Wan-Move | Wan2.1-I2V-14B | 2000K | **12.2** | **83.5** | **17.8** | **0.64** | **2.6** |

Table 9: Large-motion and out-of-distribution-motion subset.

| Subset | Method | FID↓ | FVD↓ | EPE↓ |
|---|---|---|---|---|
| Large | Tora | 29.1 | 126.3 | 4.3 |
| | MagicMotion | 24.6 | 119.3 | 4.1 |
| | Wan-Move | **14.5** | **86.6** | **3.0** |
| OOD | Tora | 28.9 | 120.2 | 4.0 |
| | MagicMotion | 23.5 | 115.7 | 3.9 |
| | Wan-Move | **13.5** | **86.0** | **2.8** |

failures (see Fig. 8). The random track embedding approach, originally proposed for pixel space representations [13], is adapted to assign randomly initialized embeddings in latent space for injecting motion guidance. While effective for rigid single-region control, this approach fails to incorporate contextual information from surrounding regions, resulting in suboptimal video quality (lower PSNR and SSIM) and stiff motion near tracked points. For example, the hand moves, but surrounding bread remains static in Fig. 8. In contrast, our proposed latent feature replication method achieves superior video quality (highest PSNR of 17.8) and precise motion control (lowest EPE of 2.6).

**Condition fusion strategy.** We compare different motion condition approaches, namely Control-Net [18] and direct concatenation (our approach). The results are presented in Table 5. Notably, simple concatenation of motion conditions with input noise achieves performance comparable to ControlNet in motion-controllable generation. Yet, ControlNet introduces significant additional modules, substantially increasing inference latency by 225 seconds over the original I2V model. In contrast, Wan-Move preserves the base model architecture and only adds a one-time trajectory extraction process, increasing just 3-second inference time.

**Number of point trajectories during training.** Table 6 evaluates the impact of the maximum number of point tracks ($N$) during training. As $N$ increases from 10 to 200, the model's motion-following capability improves progressively, evidenced by the decreasing EPE. The optimal performance, in terms of both structural similarity (SSIM) and EPE, is achieved at $N$=200. However, further increasing the number of point tracks leads to a rise in EPE. This can be attributed to the mismatch between the dense point tracks in the training and the sparse point tracks during evaluation.

**Number of point tracks during inference.** Table 12 ablates the performance of Wan-Move across varying numbers of point trajectories over MoveBench. As the number of tracks increases, EPE drops significantly, indicating better motion guidance and enhanced temporal coherence. When reaching the maximum number of point trajectories extracted by CoTracker, Wan-Move achieves the lowest EPE of 1.1. Though it is trained with at most 200 tracks, the model shows strong generalization capability. Notably,

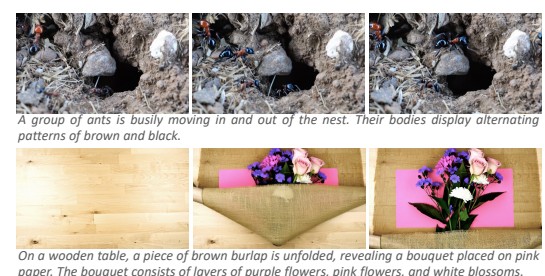

A group of ants is busily moving in and out of the nest. Their bodies display alternating patterns of brown and black.

On a wooden table, a piece of brown burlap is unfolded, revealing a bouquet placed on pink paper. The bouquet consists of layers of purple flowers, pink flowers, and white blossoms.

Figure 9: I2V results of Wan-Move (no point tracks).

naive I2V inference (with no point tracks) yields PSNR and SSIM scores comparable to motion-controlled generation, confirming that our model strongly retains its inherent I2V quality. Naive I2V samples generated by Wan-Move are presented in Fig. 9.

**Backbones and data scale.** In pursuit of the best generation quality, we initially train Wan-Move with a large-scale dataset and a strong backbone. To ensure a fair comparison with the leading approaches MagicMotion and Tora, we align Wan-Move's backbone and training data scale with them. This yields two variants, *i.e.*, Wan-Move-Cog-23K and Wan-Move-Cog-630K, which are trained on 23K and 630K data samples respectively, using CogVideoX1.5-5B-I2V [47] as backbone. The

detailed comparison on MoveBench is shown in Table 8. Under the same backbone and data scale, Wan-Move still outperforms these two powerful methods.

**Evaluation on large-motion and out-of-distribution motion scenarios.** To further verify the model generalizability, we curate subsets from MoveBench containing high-amplitude and out-of-distribution motion control cases. For each video, its motion amplitude score is computed as the average of the top 2% largest optical flow values extracted by RAFT [75]. The top 20% highest-score videos are selected as large-motion videos. Besides, we manually curate 50 uncommon motion cases as out-of-distribution subset, including complex foreground–background interactions, objects moving out of frame, and rare camera motions. Evaluation results on these challenging examples are shown in Table 9. Notably, Wan-Move consistently outperforms two leading baselines, with performance gaps further widening under these difficult condition. In addition, Wan-Move's performance only marginally drops compared to its results on the full benchmark, demonstrating its robustness.

## 5.4 Motion Control Applications

As point trajectories can flexibly represent various types of motion, Wan-Move supports a wide range of motion control applications, as showcased in Fig. 1. First, rows 1–2 show object control using single or multiple point trajectories. For camera control (row 3), we can either drag background elements directly or follow the approach of work [13]. The latter estimates a point cloud from a monocular depth predictor [76], projects it along a camera pose trajectory, and applies z-buffering to obtain camera-aligned 2D trajectories. Following work [13], we perform primitive-level control by rotating a virtual sphere to generate projected 2D trajectories for globe motion (row 4). In row 5, we enable motion transfer by applying trajectories extracted from one video to update the condition features of a different image. Row 6 shows 3D rotation control by estimating depth-based positions, applying a rotation, and projecting the results to 2D. We refer readers to the supplementary file for more visualizations and full videos.

## 6 Conclusion and Discussion

We propose Wan-Move, a simple and scalable framework for precise motion control in video generation. It represents motion with point trajectories and transfers them into latent coordinates through spatial mapping, requiring no extra motion encoder. We then inject trajectory guidance into first-frame condition features via latent feature replication, achieving effective motion control without architectural changes. For rigorous evaluation, we further present MoveBench, a comprehensive and well-curated benchmark featuring diverse content categories with hybrid-verified annotations. Extensive experiments on MoveBench and public datasets show that Wan-Move generates high-quality, long-duration (5s, 480p) videos with motion controllability on par with commercial tools like Kling 1.5 Pro's Motion Brush. We believe our open-sourced solution offers an efficient path to scale motion-controllable video generation and will empower a wide range of creators.

**Limitations and broader impacts.** Wan-Move uses point trajectories to guide motion, which can be unreliable when tracks are missing due to occlusion. While we observe that short-term occlusions can be recovered once the point reappears, showing a degree of generalization, prolonged absence may lead to loss of control (see Appendix). As with other generative models, Wan-Move carries dual-use potential. Its ability to produce realistic, controllable videos can benefit creative industries, education, and simulation, but also risks misuse for generating misleading or harmful content.

# 7 Acknowledgment

This work was supported by the National Natural Science Foundation of China (Grant No. 62576191).

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

# Wan-Move: Motion-controllable Video Generation via Latent Trajectory Guidance

## *Supplementary Material*

## Contents

## 8 Implementation Details

### 8.1 Training Data Details

Table 10 presents the composition of the filtered training datasets, which are sourced from Panda70M [77], Pixabay [78], Pexels [70], and YouTube. YouTube videos are independently collected for this study. To prevent data leakage, the videos from Pexels are strictly separated from those in the proposed MoveBench.

All videos for training are captioned using Qwen2.5-VL [79], with the prompt structure illustrated in Fig. 10. This prompt emphasizes motion and camera attributes while preserving the fundamental scene descriptions, ensuring that the model semantically understands the context and generates physically plausible motions. The same captioning prompt is applied to the videos in MoveBench.

### 8.2 MoveBench Construction Details

**Video content clustering.** Following the initial filtering stage, we conduct a rigorous content clustering process to ensure broad scenario coverage in our benchmark. Specifically, we sample 16 frames per filtered video and compute the average of their SigLip [67] features. Using k-means clustering, we group these features into 54 distinct content categories. Each category label, *e.g.*, Tennis, is then automatically captioned using Qwen2.5-VL [79]. Finally, we manually select the 15–25 most representative videos per category to maintain a balance of diversity.

**Interactive labeling.** Existing models often fail to accurately identify representative motion regions in videos, as the most prominent motion may not be optimal, and many motions terminate prematurely.

Table 10: The statistics of the training datasets.

| Dataset source | Number | Captioner |
|---|---|---|
| Panda70M [77] | 0.56M | Qwen2.5-VL |
| Pixabay [78] | 0.42M | Qwen2.5-VL |
| Pexels [70] | 0.25M | Qwen2.5-VL |
| YouTube | 0.75M | Qwen2.5-VL |

Figure 10: Prompt for video caption.

**Video Caption Prompt**

VIDEO_PROMPT = "Please describe the video in a concise and natural paragraph. Your description should follow these rules:\n"\
"a) Focus primarily on the motion and behavior of main subjects in the video, such as people or animals. Describe their actions in chronological order.\n"\
"b) Briefly describe the appearance and number of these subjects, including details like color, size, and orientation.\n"\
"c) Mention spatial relationships between subjects if relevant (e.g., in front of, to the left of, etc.).\n"\
"d) You should describe the camera perspective and movement at the end of the description, including the shooting angle (e.g., top-down, frontal, side view) and camera motion (e.g., pan left, zoom in, dolly out, slight shake), especially if they contribute to the perception of motion.\n"\
"e) Briefly describe the background or scene, but keep it minimal unless it's important for understanding the motion.\n"\
"f) Do not include text recognition, named characters, or style analysis (e.g., realistic, animated), unless they are essential for understanding the motion.\n"\
"Keep your description concise and fluent, ideally within 2–5 sentences. Your description: "

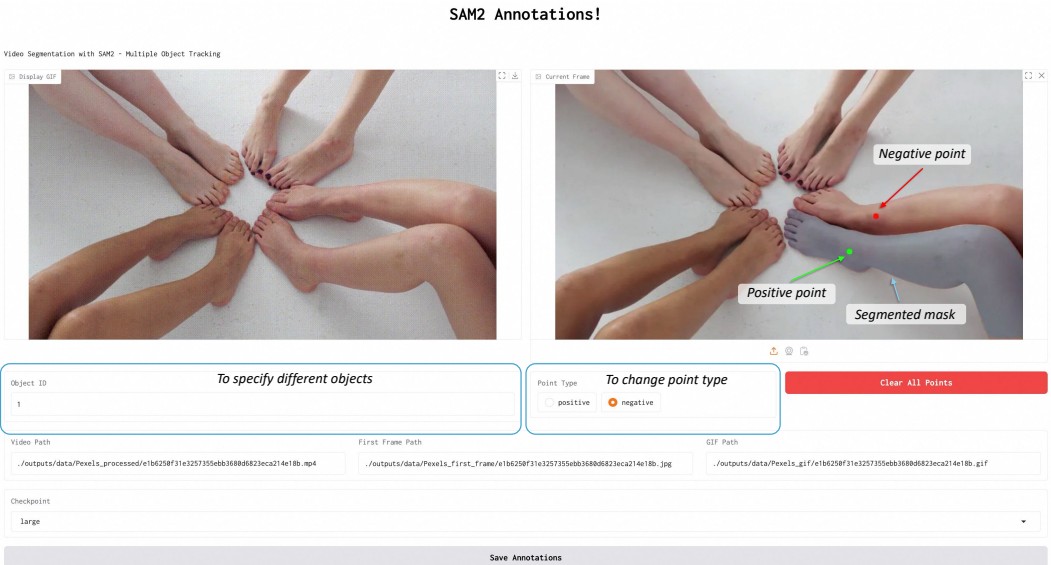

Figure 11: The interactive annotation interface displays the video (left) and its first frame (right). Users click green positive points to specify the start point of a motion trajectory, and red negative points to exclude irrelevant regions if needed. SAM segments the mask of moving objects & regions for user review. To annotate multiple motion trajectories, users must assign different object IDs.

To facilitate precise annotation of motion regions, we introduce an interactive labeling interface (Fig. 11) for selecting the initial motion point and its corresponding mask in the first frame. Annotators begin by selecting a target point in the initial frame, prompting SAM [23] to generate a preliminary segmentation mask. If the mask extends beyond the desired area, negative points can be added to exclude irrelevant regions. This method effectively isolates articulated motions or small objects. For subsequent frames, point trajectories are automatically extracted using CoTracker [68].

## 8.3 Training and Inference Configuration

As illustrated in Sec. 5.1 of the main paper, we employ Wan-I2V-14B [19] as the base I2V model. During training, both the DiT and umT5 components of Wan are wrapped with Fully Sharded Data Parallel (FSDP) [80], with parameters cast to torch.bfloat16 for memory efficiency. The training employs the AdamW optimizer [81] with a weight decay of 1e-3 and a base learning rate of 5e-6. The first 2,000 steps are used for linear warm-up to enable a smooth transition from the initial I2V generation (corresponding to 0 point trajectories) to motion-controllable video generation. We adopt flow matching objective for optimization, where the number of time sampling steps is set to 1,000 during training. To enable large-scale training with long sequences (e.g., 5s video clip), we adopt the Ulysses sequence parallelism strategy [82] following Wan, setting the sequence parallel size to 4. We train our model using 64 NVIDIA A100 GPUs, with each GPU processing a quarter of

Table 11: Impact of feature replication strategies when multiple motion trajectories overlap.

| strategy | FID↓ | FVD↓ | PSNR↑ | SSIM↑ | EPE↓ |
|---|---|---|---|---|---|
| Average | 13.1 | 83.4 | 17,5 | 0.63 | 2.7 |
| Random | 12.2 | 83.5 | 17.8 | 0.64 | 2.6 |

Table 12: Impact of using a dense-to-sampling training strategy.

| Strategy | FID↓ | FVD↓ | PSNR↑ | SSIM↑ | EPE↓ |
|---|---|---|---|---|---|
| Dense-to-Sampling | 12.9 | 84.2 | 17.5 | 0.62 | 2.6 |
| Sampling | 12.2 | 83.5 | 17.8 | 0.64 | 2.6 |

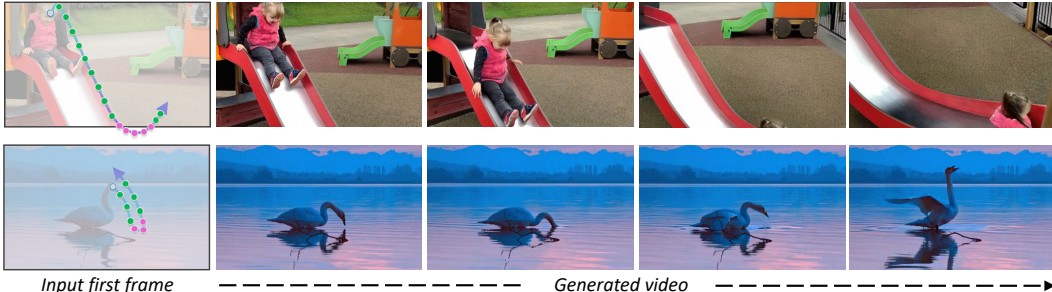

*Input first frame* — — — — — — — — — — *Generated video* — — — — — — — — — — ▶

Figure 12: Wan-Move generalizes to continue controlling motion trajectories when they temporarily disappears. Green circles indicate visible segments, while red circles mark invisible segments, *e.g.*, occluded or out-of-frame parts.

the sequence length, for a total of 30,000 steps. During inference, we follow Wan's sampling scheme with 50 sampling steps.

# 9 Additional Experiments

## 9.1 Choice of Feature Replication Strategies under Trajectory Overlap

We analyze the impact of feature replication strategies in cases of trajectory overlap. The results, presented in Table 11, demonstrate that randomly selecting a single trajectory's first-frame feature for replication when multiple trajectories coincide yields superior video quality and motion control. This is evidenced by lower FVD and EPE compared to feature averaging. We hypothesize that averaging features from overlapping trajectories leads to information loss, thereby degrading performance.

## 9.2 Choice of Different Training Strategies

This subsection evaluates the performance differences between dense-to-sampling and direct sampling training strategies, as presented in Table 12. Prior work [14, 15, 2, 53] commonly employs a two-stage dense-to-sampling pipeline, where the first stage uses dense motion trajectories to enhance motion control followed by sparse trajectories in the second stage. However, we find that our model, trained with randomly sampling of 1-200 points (as refer to Sec. 3.3 in the main paper), achieves comparable EPE and lower FVD compared to the two-stage approach. These results demonstrate that our method provides generalization capability in point trajectory numbers while simplifying the training process. This generalization ability is also verified in Table 12 of the main paper.

## 9.3 Model Performance Under Trajectory Disappearance

Fig. 12 illustrates our model's capability to generate motion-coherent videos when handling temporarily invisible trajectories. The Wan-Move maintains stable generation quality in these challenging scenarios, which we attribute to both the presence of similar cases in the training data and the model's inherent generalization capacity.

## 9.4 Failure Cases

This subsection analyzes and visualizes three primary failure modes of Wan-Move, as illustrated in Fig. 13. First, control degradation occurs when motion trajectories remain invisible for extended durations, causing the model to lose conditional guidance. Second, performance deteriorates in visually complex scenes with multiple interacting objects in crowded environments. Third, implausible

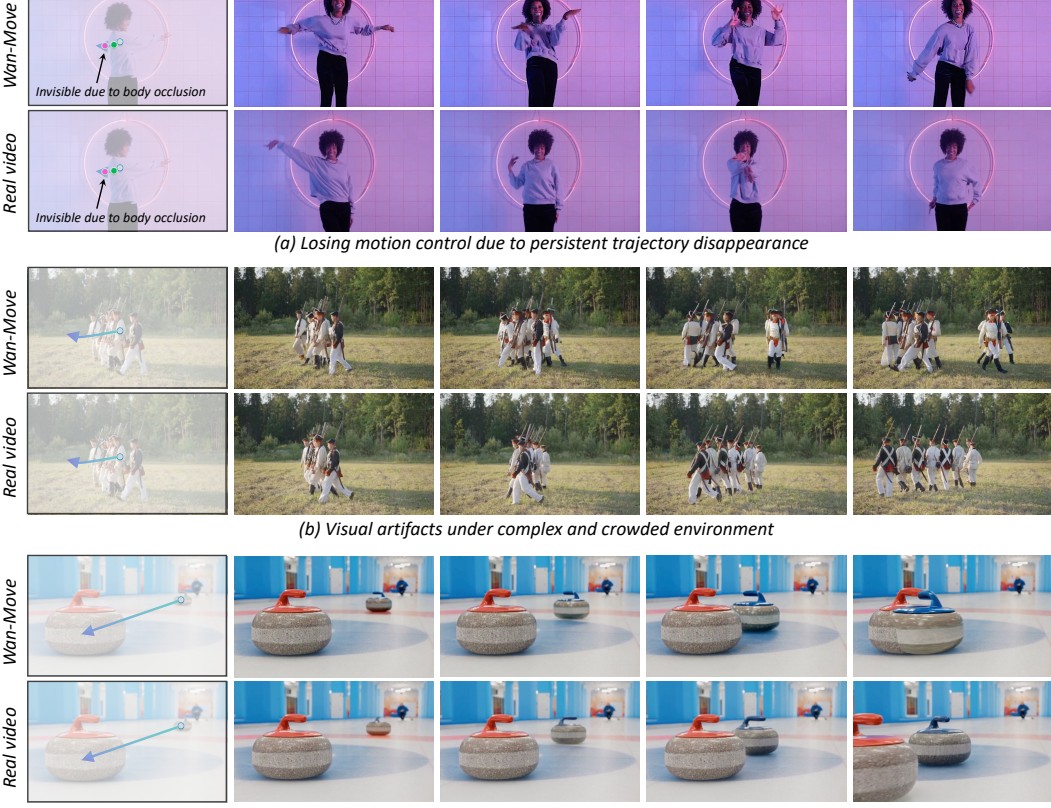

Figure 13: Three primary failure modes of Wan-Move. (a) Loss of motion control due to persistent trajectory disappearance; (b) Visual artifacts in overly complex, crowded environments; and (c) motion outputs that violate rigorous physical laws.

motion trajectories that violate fundamental physical laws result in out-of-distribution predictions. Furthermore, erroneous tracking points identified by CoTracker [68] may compound these failure modes.

## 10 Qualitative Visualizations

### 10.1 More Qualitative Comparisons

We present additional qualitative comparisons with state-of-the-art academic [14] and commercial [4] approaches, as shown in Fig. 14.

### 10.2 More Camera Control Results

As demonstrated in Fig. 15, Wan-Move enables camera control. This can be accomplished, following the work [13], by estimating a point cloud using a monocular depth predictor [76], projecting it along a predefined camera trajectory, and applying z-buffering to derive occlusion flags and camera-aligned 2D trajectories.

### 10.3 More Motion Transfer Results

This subsection presents dense motion transfer visualizations generated by Wan-Move using dense point trajectories (1,024 in our implementation). As illustrated in Fig. 16, Wan-Move achieves nearly identical appearance quality and motion alignment compared to the original videos given dense

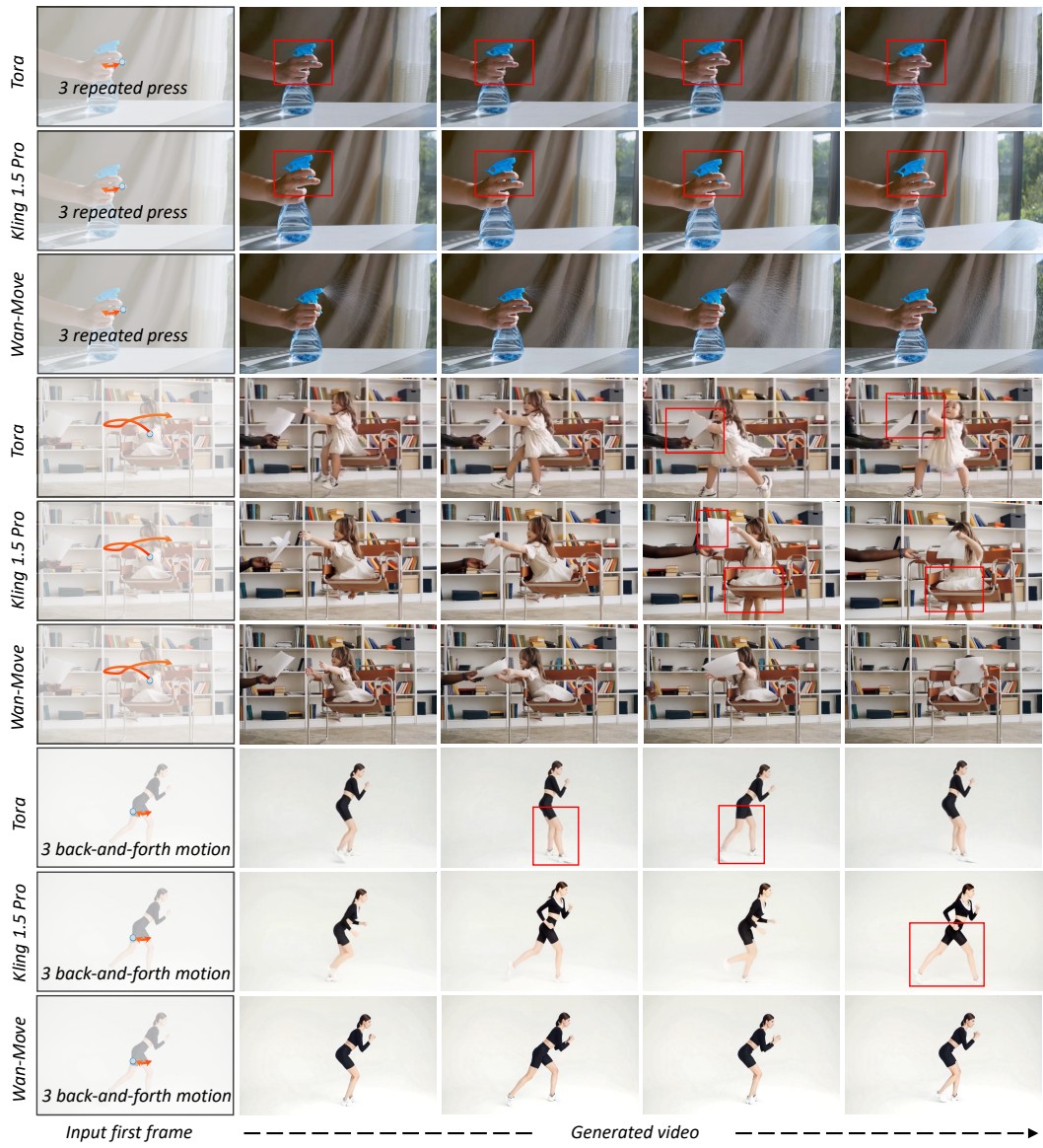

Figure 14: Additional qualitative comparisons with Tora [14] and the commercial model Kling 1.5 Pro [4]. Wan-Move demonstrates superior motion accuracy and visual quality. Major motion control failures or visual artifacts are denoted with red boxes.

trajectory conditions and the same first frame. Moreover, Wan-Move also enables video editing by copying the motion while using an additional image editing model to modify the content in the first frame, maintaining the original video's motion trajectories, as shown in Fig. 17.

## 10.4 More 3D Rotation Results

As illustrated in Fig. 18, Wan-Move additionally supports 3D object rotation. This capability is realized by first estimating depth-based 3D positions, applying a rotational transformation, and then reprojecting the results into 2D trajectories. These trajectories subsequently serve as conditioning inputs for our model to rotate the objects in videos.

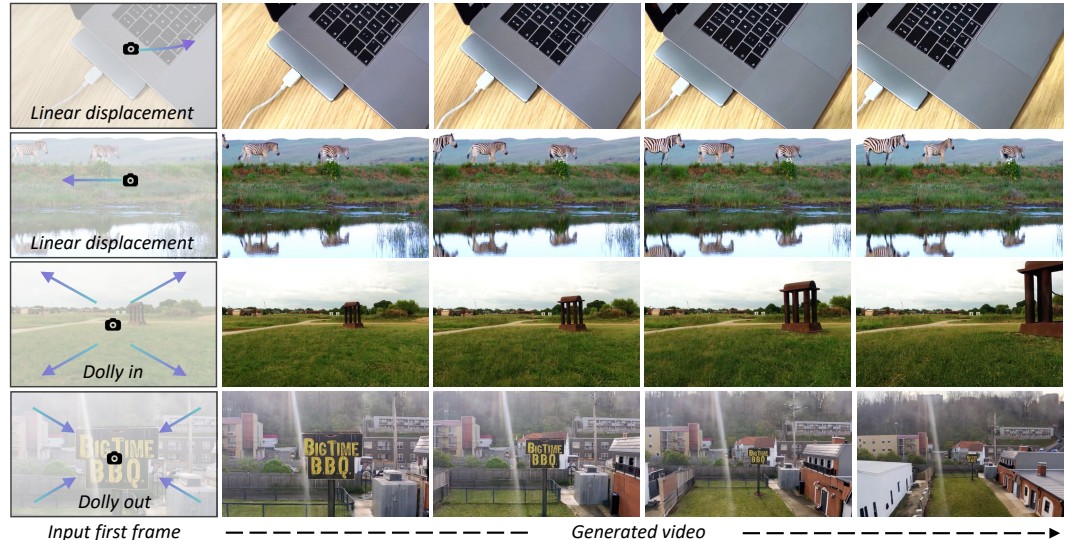

Figure 15: Wan-Move enables effective and flexible camera control through different point trajectories, such as linear displacement, dolly in, and dolly out.

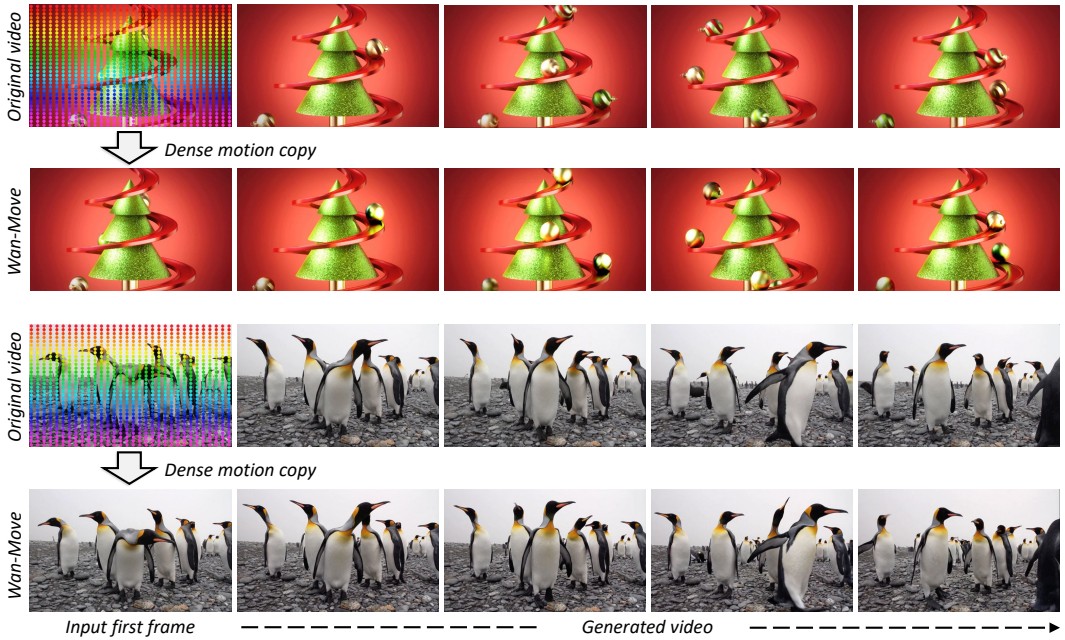

Figure 16: Wan-Move enables accurate video motion copy using dense point trajectories (*e.g.*, 1024 points). The synthesized video preserves high fidelity in both appearance and object-level motion alignment with the original video, even under complex environmental conditions.

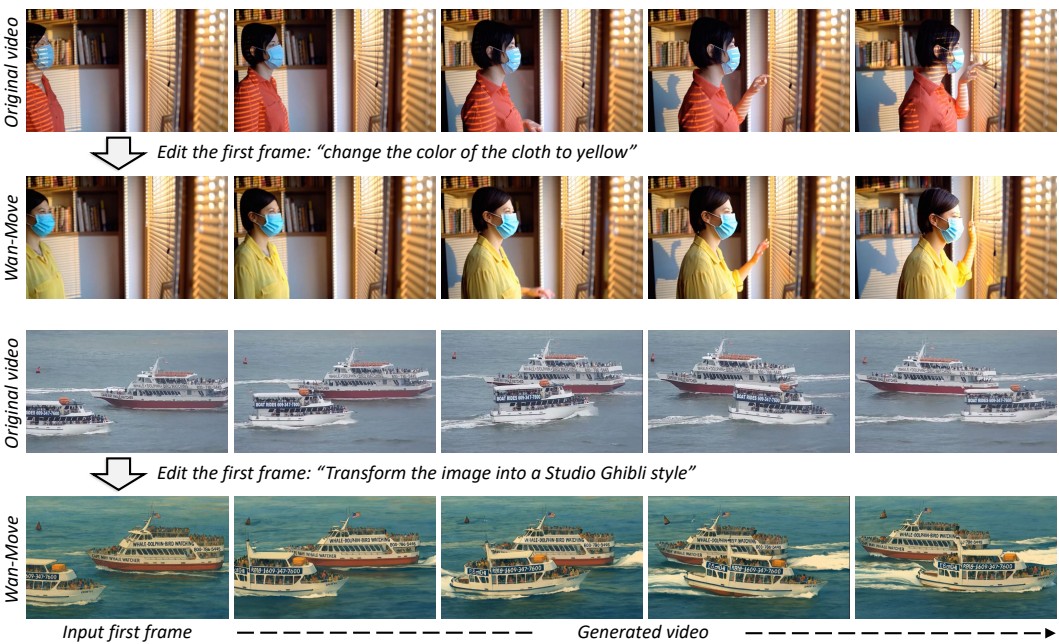

Figure 17: Wan-Move enables video editing through motion copy and additional image editing models. It first applies the image editing model (*e.g.*, ControlNet [18], GPT-4o [83]) to modify the style or content of the first frame, then uses the original video's motion trajectories to animate the edited image frame.

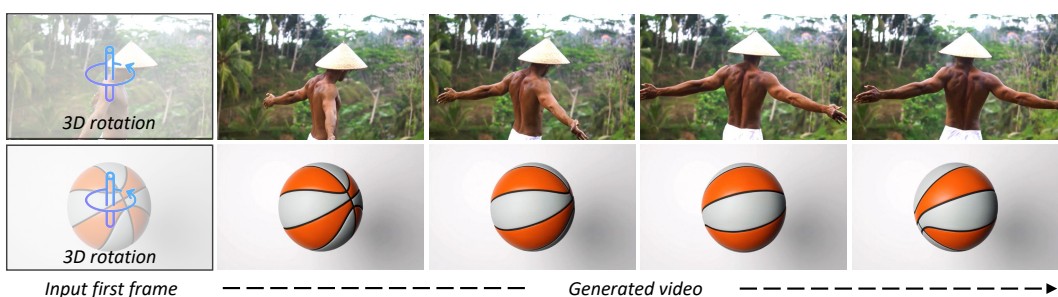

Figure 18: Wan-Move enables object 3D rotation by estimating depth-based positions, applying a rotation, and projecting the results to 2D.