# OpenReview forum: "Wan-Move: Motion-controllable Video Generation via Latent Trajectory Guidance"
_NeurIPS.cc/2025/Conference — NeurIPS 2025 poster_

### Official Review · Reviewer_TUep · 2025-06-25

**Clarity:** 3
**Significance:** 3
**Originality:** 2
**Rating:** 5
**Confidence:** 5

**Summary:**

The paper proposes Mover, a method for repurposing a pre-trained video generation model to enable motion-controllable generation. Mover achieves this by conditioning the pre-trained backbone on a sequence of latent frames constructed as follows: the first frame contains the complete latent representation of the first image in the video, while the remaining latent frames are initially zeroed out and then partially filled with features from the first frame along sparsely sampled ground truth motion trajectories. At test time, this allows the model to condition the generation on a sparse set of motion trajectories. Extensive experiments show that Mover outperforms previous methods in terms of video quality, motion accuracy, and human preference. In addition, the authors introduce MoverBench, a curated benchmark designed specifically for evaluating motion-controllable video generation.

**Questions:**

1) Could the authors include a discussion on how the control in Mover differs from [1]? The baselines in the ablations seem too weak. Have the authors tried using external visual encoders instead of the native encoder or random track embeddings?

2) Could the authors discuss how the magnitude of the control affects the accuracy of the model following the specified trajectories?

3) The control signal is applied at a low-resolution feature level. Could the authors provide insight into how this impacts fine-grained control, especially for small or subtle motions?

**Ethical Concerns:**

["NO or VERY MINOR ethics concerns only"]

**Final Justification:**

The authors did a good job during the rebuttal and addressed most of my concerns. The proposed method to cast a pretrained video model to a trajectory-controlled one is simple, clearly presented, and demonstrates good empirical results.

**Limitations:**

yes

**Paper Formatting Concerns:**

I have noticed no major formatting issues.

**Quality:**

3

**Strengths And Weaknesses:**

## Strengths
1) The paper is well-written, easy to follow, and presents the method clearly.
2) It includes sufficient implementation details to support reproducibility.
3) The experiments are comprehensive, and the quantitative results are convincing.

## Weaknesses
1) The core idea of conditioning video generation on sparse moving visual features is not novel. Prior works [1, 2] have employed similar approaches to condition the generation on motion-trajectories. Notably, these methods leverage external feature encoders that are robust to small appearance changes, eliminating the need for ground truth trajectories and allowing flexibility during inference. Although Mover demonstrates improved controllability and applies the technique to a general-purpose video model with stronger quantitative results, a thorough discussion of these related works would help contextualize the contribution.
2) The qualitative results are less convincing. The magnitude of motion control appears limited, and the number of qualitative examples is small, raising questions about the method's performance under larger and more out of distribution control inputs.
3) One limitation of the proposed control format is its low resolution. Although the tracks themselves can be of high-resolution, they are applied to low-resolution features. This may restrict the granularity of the proposed control.

[1] Hassan et. al, Gem: A generalizable ego-vision multimodal world model for fine-grained ego-motion, object dynamics, and scene composition control. CVPR 25

[2] Davtyan et. al, CAGE: Unsupervised Visual Composition and Animation for Controllable Video Generation. AAAI 25

---

> ### Author Rebuttal · Authors · 2025-07-31
>
> Dear Reviewer TUep,
>
> Thank you for the constructive comments. We are glad that you found our implementation details sufficient for reproducibility. We will also open-source our code and model.
>
> **Q1: A thorough discussion of related works [1,2] would help contextualize the contribution.**
>
> A1: Thanks for your constructive feedback. We are happy to clarify their differences and incorporate this discussion into the revised Related Work section.
>
> Works [1,2] and Mover are simlar in that they condition the generation on latent motion trajectories. However, they differ in the methodology and task. **(i) Methodology difference:** Works [1,2] rely on pretrained DINOv2 features to transfer object representations across frames, yet both papers highlight DINOv2’s limitations in video generation. First, DINOv2 excels at high-level semantic encoding but lacks fine-grained object details. Thus, GEM [1] employs additional identity embeddings to distinguish objects and train an ObjectNet to bridge the domain gap between DINOv2 and the
> UNet’s feature space. Second, CAGE [2] observes that DINOv2 features inherently retain spatial positioning. This makes it hard to disentangle object identity from the original location, finally reducing controllability. Moreover, the 14 $\times$ 14 patch size in DINOv2 may restrict the granularity of the proposed control. In contrast, our approach mitigates these limitations without relying on any auxiliary modules (e.g., identity embeddings). **(ii) Task difference:** As you mentioned, we focus on general-purpose video generation, supporting a significantly broader range of motion control tasks compared to [1-2].
>
> **Q2: The qualitative results contain a limited number of large-motion examples, raising questions about the model performance under larger and more out-of-distribution motion conditions.**
>
> A2: Our visualization results include a number of large-motion cases, such as boxing (supplementary video: 10–15s) and aerobics (47–52s). The supplementary video also shows that Mover performs noticeably better than compared methods in high-amplitude motion scenes (47s-1:11). This capability is supported by our data and benchmark construction process, where we specifically retained a wide range of large-motion cases using expert models and motion-based filtering (Lines 165–170 and 204–207).
>
> To further address your concern, we curate subsets from MoverBench containing high-amplitude and out-of-distribution motion control cases. Evaluation results on these challenging examples are shown in **Tables R5** and **R6**. Notably, Mover consistently outperforms two leading baselines [3,4], with performance gaps further widening under these difficult condition. In addition, Mover’s performance only marginally drops compared to its results on the full benchmark, demonstrating its robustness. Due to rebuttal guidelines, we cannot include additional visual examples here, but we will provide more such cases in our upcoming project page and include these two tables for clarity.
>
> **Table R5:** Comparisons on MoverBench large-motion subset. For each video, its motion amplitude score is computed as the average of the top 2% largest optical flow values extracted by RAFT [5]. The top 20% highest-score videos are selected.
> | Method     | FID  $\downarrow$ | FVD  $\downarrow$ | EPE  $\downarrow$ |
> |------------------|:-------:|:-------:|:-------:|
> | Tora [3]  | 29.1  | 126.3   | 4.3     |
> | MagicMotion [4]   | 24.6   | 119.3  | 4.1    |
> | Mover  | **14.5**   | **86.6**   | **3.0**    |
>
> **Table R6:** Comparisons on MoverBench out-of-distribution motion subset. We manually curated 50 uncommon motion cases, including complex foreground–background interactions, objects moving out of frame, and rare camera motions.
> | Method    | FID  $\downarrow$ | FVD  $\downarrow$ | EPE  $\downarrow$ |
> |------------------|:-------:|:-------:|:-------:|
> | Tora [3]  | 28.9   | 120.2   | 4.0    |
> | MagicMotion [4]   | 23.5   | 115.7   | 3.9    |
> | Mover   | **13.5**   | **86.0**   | **2.8**  |
>
>
> **Q3: One limitation of the proposed control format is its low resolution. Could the authors provide insight into how this impacts fine-grained control?**
>
> A3: We use a VAE encoder to compress motion trajectories with a spatial downsampling ratio of 8, which is a relatively low compression. It means that when the trajectory displacement exceeds approximately 8 pixels, the model can theoretically distinguish them. Even when the motion amplitude is very small (a particularly demanding case), the model may still infer the intended motion by leveraging the text prompt (e.g., "slightly move to the left") and the contextual motion patterns from nearby regions.
>
> To further address your concern, we conduct a targeted evaluation. We sample 100 videos from MoverBench and, for each video, select one point trajectory with total displacement falling into one of four ranges: <8 pixels, 8–16 pixels, 16–32 pixels, and >32 pixels. We perform motion control using the selected trajectories and evaluate the control precision by end-point error (EPE). As shown in **Table R7**, Mover maintains reasonable control accuracy even for very small motions, demonstrating fine-grained motion control capability.
>
> **Table R7:** Comparisons of motion control performance at varying motion amplitudes.
> | Trajectory displacement |  <8 pixels | 8–16 pixels | 16–32 pixels | >32 pixels |
> |------------------|:-------:|:-------:|:-------:|:-------:|
> | **EPE**   | 3.0   |  2.6  | 2.5    |2.5    |
>
> Moreover, there are two potential solutions to further alleviate the resolution limitation. **(i)** We have already trained Mover based on the I2V-14B-720P backbone, increasing the video resolution from 480P to 720P and enabling finer motion granularity. We are happy to release this model. **(ii)** Another direction is to incorporate explicit pixel-level position embedding into our feature replication mechanism. We leave this as a promising future work.
>
> **Q4: Have the authors tried using external visual encoders instead of the native encoder or random track embedding?**
>
> A4: Thanks for the suggestion. During the rebuttal phase, we evaluate two external visual encoder designs. **(i)** Mover-Motion_Encoder: we add learnable convolutional layers (like Tora’s 'Trajectory Extractor') after the feature replication step to further encode the motion condition. **(ii)** Mover-DINOv2: We extract features from a pretrained DINOv2 model and resize them to match the latent feature map for conditioning. As shown in **Table R8**, both variants underperform compared to our approach. Mover-Motion_Encoder increases optimization complexity and degrades performance. Mover-DINOv2 yields the weakest results. It aligns with prior findings (see Q1 & A1) that DINOv2 features, while semantically rich, introduce limitations for motion-controllable video generation. We are happy to incorporate this experiment for a more complete ablation in the revised paper.
>
> **Table R8:** Comparisons of visual encoders on MoverBench.
> | Approach    | FID  $\downarrow$ | FVD  $\downarrow$ | EPE  $\downarrow$ |
> |------------------|:-------:|:-------:|:-------:|
> | Mover-DINOv2   | 16.6   | 97.2   | 3.5    |
> | Mover-Motion_Encoder       | 14.5   | 88.2   | 3.0   |
> | Mover (full fine-tuning)   | **12.2**   | **83.5**   | **2.6**    |
>
> ****
> **References**
>
> [1] Hassan, et al. Gem: A Generalizable Ego-vision Multimodal World Model for Fine-grained Ego-motion, Object Dynamics, and Scene Composition Control. In CVPR 2025.
>
> [2] Davtyan, et al. CAGE: Unsupervised Visual Composition and Animation for Controllable Video Generation. In AAAI 2025.
>
> [3] Zhang, et al. Tora: Trajectory-oriented Diffusion Transformer for Video Generation. In CVPR 2025.
>
> [4] Li, et al. MagicMotion: Controllable Video Generation with Dense-to-Sparse Trajectory Guidance. In ICCV 2025.
>
> [5] Teed, et al. RAFT: Recurrent All-Pairs Field Transforms for Optical Flow. In ECCV 2020.

---

> > ### Comment · Reviewer_TUep · 2025-08-04
> > **Response to the Rebuttal**
> >
> > Thank you for the rebuttal. I appreaciate the detailed clarifications. I have also read the other reviews and have no further questions to the authors. Therefore I plan to raise my rating to Accept.

---

> > > ### Author Response · Authors · 2025-08-04
> > > **Author Response to Reviewer TUep**
> > >
> > > Dear Reviewer TUep,
> > >
> > > Many thanks for your helpful comments and supportive words. Your suggestions greatly help us refine our work.

---

### Official Review · Reviewer_wpp4 · 2025-07-01

**Clarity:** 3
**Significance:** 2
**Originality:** 2
**Rating:** 4
**Confidence:** 4

**Summary:**

The paper introduces Mover, a framework for precise motion control in video generative models. It leverages dense point trajectories to enable fine-grained control over scene motion. These trajectories guide the propagation of features from the first frame, generating motion-aware latent conditions. Mover integrates seamlessly into existing models without requiring architectural changes or additional encoders. Additionally, the paper presents MoverBench, a new benchmark for evaluating motion-controllable video generation. Experiments and user studies demonstrate that Mover outperforms existing methods.

**Questions:**

Please see weakness section.

**Ethical Concerns:**

["NO or VERY MINOR ethics concerns only"]

**Final Justification:**

I have carefully reviewed the authors' response as well as the comments from the other reviewers. Most of my concerns have been addressed. Although I still feel that the technical contribution of the paper could be further improved, I will increase my score to borderline accept.

**Limitations:**

yes

**Quality:**

2

**Strengths And Weaknesses:**

Strengths:
1. The technical description is concise and easy to understand.
2. The use of point trajectories as control signals makes the framework user-friendly.

Weaknesses:
1. Limited Technical Novelty: The core technique of the paper—latent feature replication along point trajectories—is conceptually similar to the widely tested pixel replication methods. Extending this idea to the latent domain offers only marginal novelty. Additionally, concatenating noisy video latents with condition inputs has also been widely adopted, such as in Wan [a]. It would be helpful if the authors could briefly clarify the distinctions between their method and related approaches like ATI [b], which also leverage VAE latent propagation.
2. Scalability Claim Lacks Sufficient Support: The paper's claim that Mover is scalable is not convincingly substantiated. Many existing methods can also be fine-tuned with minimal architectural changes and computational overhead to achieve similar capabilities. While the paper compares inference speed with ControlNet-style branching methods, a more comprehensive evaluation is needed—including training cost across different backend models, convergence speed, and resulting model performance.
3. Inconsistent Baseline Models: The evaluation lacks rigor due to differing backend models used across comparisons. For instance, Tora is based on CogVideoX, while Mover is tested on Wan-I2V-14B. Given the performance gap between these two models, it is difficult to draw fair conclusions about Mover's advantages solely from these comparisons.
4. It remains unclear how Mover handles ambiguous control signals. Specifically, when a control point and its associated trajectory are provided, it is not evident how Mover distinguishes whether the signal is intended to control local object deformation, global object motion (translation or rotation), camera movement, or background object motion. Further clarification on this aspect would strengthen the paper.

[a] Wan, Team, et al. "Wan: Open and advanced large-scale video generative models." arXiv preprint arXiv:2503.20314 (2025).

[b] Wang, Angtian, et al. "ATI: Any Trajectory Instruction for Controllable Video Generation." arXiv preprint arXiv:2505.22944 (2025).

---

> ### Author Rebuttal · Authors · 2025-07-31
>
> Dear Reviewer wpp4,
>
> Thanks for the constructive comments. Please find our response to the specific points below.
>
>
> **Q1: Latent feature replication is conceptually similar to the pixel replication methods.**
>
> A1: We would like to clarify that latent feature replication is non-trivial. Compared to pixel-level replication, it offers two advantages crucial for overall performance. **(i)** Pixel-replicated videos produces out-of-distribution visual inputs for pretrained VAEs, typically requiring auxiliary encoder networks for enhanced performance (e.g., Tora [1], discussed in Lines 93-97). This complicates the fine-tuning of video-generation backbones, which contrasts with our core insight. **(ii)** Individual pixels lack semantic and contextual information (referred in Lines 34-35). Directly copying them hinders motion pattern alignment across surrounding regions. In contrast, latent features carry richer contextual cues, enabling more coherent motion conditions.
>
> These advantages were rigorously validated in Table 4 and Figure 8 of the main paper, both quantitative and qualitatively. Specifically, our 'latent feature replication' approach demonstrates significant performance gains over the 'pixel replication', lowering FVD by 7.5.
>
>
>
> **Q2: Concatenating noisy video latents with condition inputs has been widely adopted such as in Wan.**
>
> A2: Yes, as clearly stated in Lines 135–138, this operation is used in prior I2V backbones such as Wan [2] and Lumiere [3] for fusing image conditions. Because this strategy is already integrated into the backbone's design, we follow the same approach to maintain compatibility and avoid modifying the model architecture. We also provide an ablation in Table 5, which confirms that the concatenation strategy works well for our task.
>
> **Q3: Clarify the distinctions between Mover and ATI.**
>
> A3: We note that ATI [4] (arXiv: May 28) was submitted after our NeurIPS 2025 submission deadline (May 15). Our work was developed independently.
>
> **Q4: Scalability claim needs more sufficient support.**
>
> A4: Thanks for your suggestion. Mover’s scalability mainly stems from preserving the base I2V model architecture. To this end, we **(i)** simplify motion control extraction by eliminating trajectory encoders like Tora’s [1], and **(ii)** avoid ControlNet-style fusion module that adds a new branch like MagicMotion’s [5]. To address your concern, we perform ablation studies to evaluate how these two designs facilitate scalability. The first compares Mover with a variant that adds a trainable motion encoder (denoted as Mover-Motion_Encoder), the second compares Mover with its ControlNet-style variant (denoted as Mover-ControlNet). Experiments are extensively conducted on two backbones, using the same data for fairness.
>
> As shown in **Table R3**, our method achieves both better performance and faster convergence compared to alternatives. Mover-Motion_Encoder increases optimization complexity and degrades performance, Mover-ControlNet converges more slowly, possibly due to using zero convolutions, as also observed in ControlNext [6]. We are happy to incorporate the tables to better clarify our scalability.
>
> **Table R3:** Scalability comparison of different approaches on two backbones.
>
> | Approach                     | Backbone       | Convergence Step $\downarrow$  | FVD $\downarrow$  | FID $\downarrow$ | EPE $\downarrow$  |
> |-----------------------------|:--------------:|:-------------------:|:------:|:------:|:------:|
> | Mover-Motion_Encoder   | Wan2.1-I2V-14B | 33000 | 13.5 | 87.8 | 2.7    |
> | Mover-ControlNet   | Wan2.1-I2V-14B | 36000  | 12.4   | 84.6   | **2.5**    |
> | Mover      |Wan2.1-I2V-14B  | **29500** | **12.2** | **83.5** | 2.6   |
> |||||||
> | Mover-Motion_Encoder   | CogVideoX-5B| 26500 | 14.5 | 88.2 | 3.0 |
> | Mover-ControlNet   | CogVideoX-5B| 29500  | **14.1** | 87.0   | 2.8 |
> | Mover      | CogVideoX-5B  |  **23000** | 14.2  | **86.6**   | **2.8** |
>
>
> *Setting illustration:* In **Table R3**, Mover-Motion_Encoder refers to that we add convolutional layers (following 'Trajectory Extractor' in Tora) after the feature replication step to further encode the motion condition. We define convergence as the point where evaluation metrics stabilize within a small range.
>
> **Q5: Mover and baseline methods use different backbones.**
>
> A5: We choose Wan as the backbone to pursue performance upper bound among open-source models. Notably, we can compare our performance with commercial model (Kling 1.5 Pro). To address your concern, we adjust Mover's backbone and training data scale to match those of two leading methods, MagicMotion and Tora. The comparisons under aligned settings are shown in **Table R4**. Under the same backbone and training data scale, Mover still outperforms these powerful methods. We are happy to include this table in our revised paper for clarity.
>
> **Table R4:** Comparison of model performance under different training setups. Mover-Cog-23K and  Mover-Cog-630K denote Mover variants under different backbones and training data scales.
>
> | Method               | Backbone         | Data Scale | FID  $\downarrow$ | FVD  $\downarrow$ | PSNR  $\uparrow$ | SSIM $\uparrow$ | EPE $\downarrow$ |
> |----------------------|:----------------:|:----------:|:-----:|:-----:|:------:|:------:|:-----:|
> | MagicMotion [5]    | CogVideoX-5B      | 23K        | 17.5  | 96.7 | 14.9  |0.56  | 3.2 |
> | Mover-Cog-23K           | CogVideoX-5B      | 23K        | **16.0** | **92.3**  | **16.8**   | **0.59**  | **2.8**   |
> |||||||||
> | Tora [1]     | CogVideoX-5B      | 630K       | 22.5  | 100.4 | 15.7  | 0.55  | 3.3  |
> | Mover-Cog-630K                    | CogVideoX-5B      | 630K       | **14.1**  | **87.3** | **17.2**   | **0.61**  | **2.8**  |
> |||||||||
> | Mover               | Wan2.1-I2V-14B   | 2000K      | **12.2** | **83.5** | **17.8** | **0.64** | **2.6**   |
>
>
> **Q6: How does Mover handle ambiguous control signals?**
>
> A6: Mover naturally addresses the ambiguity issue through two key mechanisms. **(i)** During inference, users can easily clarify intended motion by specifying more detailed point trajectories. For example, placing multiple points on an object with similar motion suggests global movement, while adding stationary points nearby helps indicate local movement. These adjustments can be made directly through the user interface.
>
> **(ii)** Text prompt provides complementary motion guidance, further resolving potential ambiguities in control signals. During training, text prompts clearly describe the moving foreground subject and background context, paired with randomly sampled point trajectories. With large-scale training on diverse aligned samples, even given a single point trajectory, the model can rely on text prompts to infer the desired motion.
>
>
> ****
> **References**
>
> [1] Zhang, et al. Tora: Trajectory-oriented Diffusion Transformer for Video Generation. In CVPR 2025.
>
> [2] Wan Team, et al. Wan: Open and Advanced Large-Scale Video Generative Models. arXiv preprint arXiv:2503.20314 (2025).
>
> [3] Bar-Tal, et al. Lumiere: A Space-Time Diffusion Model for Video Generation. In SIGGRAPH Asia 2024 Conference.
>
> [4] Wang, et al. ATI: Any Trajectory Instruction for Controllable Video Generation. arXiv preprint arXiv:2505.22944 (2025).
>
> [5] Li, et al. MagicMotion: Controllable Video Generation with Dense-to-Sparse
> Trajectory Guidance. In ICCV 2025.
>
> [6] Peng, et al. ControlNeXt: Powerful and Efficient Control for Image and Video Generation. arXiv preprint arXiv:2408.06070 (2024).

---

> > ### Comment · Reviewer_wpp4 · 2025-08-06
> >
> > I have carefully reviewed the authors' response as well as the comments from the other reviewers. Most of my concerns have been addressed. Although I still feel that the technical contribution of the paper could be further improved, I will increase my score to borderline accept.

---

> > > ### Author Response · Authors · 2025-08-07
> > > **Author Response to Reviewer wpp4**
> > >
> > > Dear Reviewer wpp4,
> > >
> > > We sincerely appreciate your positive feedback and for raising the score. Your suggestions have helped us further refine the paper and clarify our technical contributions. In the revised version, we will include the additional experiments as discussed to better support the model’s capability. We are also happy to release our model and code to facilitate future research in motion-controllable video generation.

---

### Official Review · Reviewer_rUr2 · 2025-07-01

**Clarity:** 3
**Significance:** 3
**Originality:** 2
**Rating:** 4
**Confidence:** 2

**Summary:**

The paper introduces Mover, a framework to add precise motion control to image-to-video (I2V) diffusion models without modifying their architectures. The key ideas are:

Latent Trajectory Guidance: Convert dense point trajectories from pixel space into the latent feature map, then replicate first-frame features along these latent paths to inject motion without auxiliary encoders (Sec. 3.2).

MoverBench: A new benchmark of 1,018 videos (5 s, 480 p) with hybrid human+SAM annotations covering 54 content categories for rigorous evaluation of motion-controllable generation (Sec. 4).

Extensive quantitative (FID, FVD, PSNR/SSIM, EPE) and human studies demonstrate that Mover outperforms recent academic methods and rivals commercial tools such as Kling 1.5 Pro’s Motion Brush.

**Questions:**

Please according to the weakness section.

**Ethical Concerns:**

["NO or VERY MINOR ethics concerns only"]

**Final Justification:**

After the rebuttal, most of my concerns are resolved, thus I will keep my positive score.

**Limitations:**

The author adequately addressed the limitations in the main paper.

**Quality:**

3

**Strengths And Weaknesses:**

**Strengths**

Comprehensive benchmark (MoverBench) with precise point and mask tracks along with detailed captions, enabling standardized evaluation of diverse motion tasks.

Strong empirical results: Significant improvements in motion accuracy (EPE) and video fidelity across single- and multi-object scenarios, supported by user studies with win rates exceeding 95% against top baselines.

**Weakness**

Using trajectories to control video generation is not in itself a particularly novel paradigm. There are several recent works employing similar mechanisms, and in essence it may not differ substantially from using optical flow, segmentation masks, or bounding boxes for control. The primary contribution of this paper appears to be the assembly of a large, trajectory‐annotated dataset and the demonstration that scaling up training yields strong results.

That said: (1) I am not a specialist in video synthesis, and different subfields sometimes value contributions in distinct ways; and (2) some related works are currently only on arXiv and could be seen as concurrent efforts. Accordingly, I would prefer to frame this review positively and defer to other reviewers’ judgments as well.

---

> ### Author Rebuttal · Authors · 2025-07-31
>
> Dear Reviewer rUr2,
>
> Thanks for your valuable feedback. We are happy to address your comments below.
>
> **Q1: Novelty of using trajectory control for video generation.**
>
> A1: We would like to clarify that using trajectories for motion control is not our novelty. Point trajectories, optical flow, and segmentation masks are all widely-used motion control forms in prior works, as explicitly discussed in the Introduction (Lines 28–36). Our method simply adopts point trajectories as the motion signal.
>
> Our novelty lies in the feature replication mechanism, which edits the image condition features directly to enable motion control. This design removes the need for training extra motion encoders or injection modules like ControlNet, leaving the base I2V model architecture unchanged. Based on this insight, we validate the method at scale, construct a well-curated benchmark for evaluation, and will release an open-source model that even rivals Kling 1.5 Pro’s commercial Motion Brush.
> Together, our method, scaled training, evaluation benchmark, and model performance form an integrated contribution to the community.
>
> We are also glad to see your recognition of the overall value of our work. We hope our insights and the released model will be helpful to both the research community and video creators.

---

> ### Comment · Reviewer_rUr2 · 2025-08-05
>
> After the rebuttal, most of my concerns are resolved, thus I will keep my positive score.

---

> > ### Author Response · Authors · 2025-08-05
> > **Author Response to Reviewer rUr2**
> >
> > Dear Reviewer rUr2,
> >
> > We appreciate your positive feedback and continued support. We're also glad to see that the contribution of our work is recognized. Thank you again for your thoughtful efforts throughout the review process.

---

### Official Review · Reviewer_b1sz · 2025-07-03

**Clarity:** 3
**Significance:** 3
**Originality:** 3
**Rating:** 5
**Confidence:** 4

**Summary:**

The paper proposes a representation for trajectory conditions and achieving such guidance in image-to-video generation. The representation basically transforms a sequence of points into the latent space and then uses the new representation to create feature maps for subsequent frames after the 1st frame. These condition features are concatenated with the noisy patches as input to the DiT backbone without having the need to modify the overall architecture or introduce new tunable modules. The authors curated a large-scale dataset for training and also a benchmark called MoverBench for evaluation. Experimental results show that Mover outperforms many previous baselines.

**Questions:**

See weaknesses

**Ethical Concerns:**

["NO or VERY MINOR ethics concerns only"]

**Final Justification:**

The authors have addressed all my concerns.

**Limitations:**

See weaknesses

**Quality:**

3

**Strengths And Weaknesses:**

(+) The paper comes up with a very smart way to represent trajectories in the latent space and then uses these representations to create conditional features to achieve motion control. This eliminates the need for an additional control network or a new tunable module, making the entire model more compact and unified.

(+) A large-scale benchmark is introduced, which seems of high quality and could benefit the community by providing a standard benchmark beyond DAVIS.

(+) Experimental results suggest superior control achieved by Mover compared to previous baselines. It also works well with multiple trajectories.

(-) While Mover saves from using an additional module, it also has the drawback of changing the whole parameter of the original DiT backbone. But with ControlNet, one gets to freeze the backbone and tune half of the parameters and use multiple trained ControlNets in a plug-and-play manner.

(-) The evaluation comparison is not fair. Mover has used a lot more high-quality videos compared to all the baselines and potentially a stronger base model (Wan-14B v.s. e.g., CogVideoX in baselines). Of course, it's never easy for this type of work to make a fair comparison due to all kinds of reasons and limitations. For example, the most comparable model is Motion Prompting, which trains a naive ControlNet with a similar number of videos as Mover -- but it is not open-sourced. However, the work needs to clarify the differences between Mover and the baselines in terms of base models and the number of training videos.

---

> ### Author Rebuttal · Authors · 2025-07-31
>
> Dear Reviewer b1sz,
>
> Thank you for the valuable comments and for recognizing the value of our method, the benchmark design, and the experimental results. We are happy to respond to each question below.
>
> **Q1: Compared to ControlNet that freezes the backbone and tunes half of the parameters, Mover changes the whole parameters of the original DiT backbone.**
>
> A1: We fully update the DiT backbone due to a better trade-off between video quality and inference cost. As shown in Table 5 of the main paper, compared to the ControlNet-based Mover variant, our full fine-tuning approach performs favorably ($\downarrow$ 0.2 FID and $\downarrow$ 1.1 FVD), while reducing test time by 22.4% and overall parameter size by over 30%. Moreover, despite updating the entire backbone, Mover effectively preserves the original image-to-video (I2V) capability of the Wan2.1-I2V backbone, as verified both quantitatively (Table 7) and qualitatively (Figure 9). This is because motion condition is occasionally dropped during training.
>
> We also agree that plug-and-play modules have their advantages. In addition to the ControlNet variant, we now train a Mover-LoRA version, which retains Mover’s design while replacing full tuning with LoRA. This version is also a plug-and-play solution and adds only 1.8% parameters. As compared in **Table R1** below, Mover (full fine-tuning) achieves the best overall performance over its two variants, i.e., Mover-ControlNet and Mover-LoRA. Notably, Mover-LoRA also consistently outperforms all academic baselines in the main paper. We are happy to release all variants to support diverse usage scenarios.
>
> **Table R1:** Comparisons of training strategies on MoverBench.
> | Approach    | FID  $\downarrow$ | FVD  $\downarrow$ | EPE  $\downarrow$ | Latency (s) $\downarrow$|
> |------------------|:-------:|:-------:|:-------:|:--------------:|
> | Mover-LoRA   | 13.9   | 85.4   | 2.7    | 765   |
> | Mover-ControlNet       | 12.4   | 84.6   | **2.5**    | 987|
> | Mover (full fine-tuning)   | **12.2**   | **83.5**   | 2.6    | **765**  |
>
>
> **Q2: Among existing methods, Motion Prompting is the most comparable in terms of training data scale, but it is not open-sourced. While a fully fair comparison is difficult, the work needs to clarify the differences in backbones and training data scale with baselines.**
>
> A2: We train Mover with large-scale data and a stronger backbone so as to pursue the best possible generation quality. As a result, Mover performs competitively with the commercial model Kling 1.5 Pro Motion Brush in human evaluations, which has not been achieved by prior academic works.
>
> We fully acknowledge your suggestion and will clarify more setting difference. Although Motion Prompting is closed-source, we have compared one of its core design with Mover fairly. Specifically, Motion Prompting formulates motion guidance using random track embeddings, while Mover adopts latent feature replication. Based on the same dataset and backbone, we directly compare the two designs in Table 4 (quantitative results) and Figure 8 (qualitative comparisons). Our approach consistently outperforms the Motion Prompting's design in video quality.
>
> Additionally, during the rebuttal phase, we adjust Mover's settings to align with two leading methods MagicMotion [1] and Tora [2], denoted as Mover-Cog-23K and Mover-Cog-630K, respectively. The detailed comparison on MoverBench is shown in **Table R2** below. Under the same backbone and training data scale, Mover still outperforms these two powerful methods. We will include this table in our revised paper for clarity.
>
> **Table R2:** Comparisons of model performance under different training setups.
>
> | Method               | Backbone         | Data Scale | FID  $\downarrow$ | FVD  $\downarrow$ | PSNR  $\uparrow$ | SSIM $\uparrow$ | EPE $\downarrow$ |
> |----------------------|:----------------:|:----------:|:-----:|:-----:|:------:|:------:|:-----:|
> | MagicMotion [1]    | CogVideoX-5B      | 23K        | 17.5  | 96.7 | 14.9  |0.56  | 3.2 |
> | Mover-Cog-23K           | CogVideoX-5B      | 23K        | **16.0** | **92.3**  | **16.8**   | **0.59**  | **2.8**   |
> |||||||||
> | Tora [2]     | CogVideoX-5B      | 630K       | 22.5  | 100.4 | 15.7  | 0.55  | 3.3  |
> | Mover-Cog-630K                    | CogVideoX-5B      | 630K       | **14.1**  | **87.3**  | **17.2**   | **0.61**  |**2.8**  |
> |||||||||
> | Mover               | Wan2.1-I2V-14B   | 2000K      | **12.2** | **83.5** | **17.8** | **0.64** | **2.6**   |
>
> ****
> **References**
>
> [1] Li, et al. MagicMotion: Controllable Video Generation with Dense-to-Sparse Trajectory Guidance. In ICCV 2025.
>
> [2] Zhang, et al. Tora: Trajectory-oriented Diffusion Transformer for Video Generation. In CVPR 2025.

---

### Comment · Area_Chair_9pZS · 2025-08-04
**Request for your feedback in light of authors' feedback**

Thank you for your valuable insights and expertise which have contributed significantly to the review process.

Following the initial review, the authors have provided a detailed rebuttal addressing the feedback and comments provided by our esteemed reviewers, including yourself. I kindly request that you take the time to carefully review the authors' rebuttal and assess its impact on your initial evaluation.

Please share your thoughts and any additional points you may have after reading the authors' rebuttal. Thank you very much!

---

### Decision · Program_Chairs · 2025-09-17

**Decision:**

Accept (poster)

**Comment:**

The paper introduces Mover, a framework for precise motion control in video generative models. It leverages dense point trajectories to enable fine-grained control over scene motion. The work is further supported by the release of MoverBench, a large-scale benchmark with precise annotations that is likely to benefit the community.

Reviewers consistently acknowledge the strengths of the paper: the method is simple yet effective, the benchmark is valuable, and empirical results show strong performance. Two reviewers gave clear accept ratings.

The main concerns raised relate to limited novelty, fairness of comparisons due to stronger base models and larger training data, and insufficient discussion of scalability and relation to recent concurrent works. However, the rebuttal addressed many of these issues, and reviewers increased or maintained their positive stance.

All reviewers reached a consensus to accept the paper.